# Mouse TRPA1 function and membrane localization are modulated by direct interactions with cholesterol

Justyna B Startek[1,2], Brett Boonen[1,2], Alejandro López-Requena[1,2], Ariel Talavera[3], Yeranddy A Alpizar[1,2], Debapriya Ghosh[1,2], Nele Van Ranst[1,2], Bernd Nilius[1,2], Thomas Voets[1,2], Karel Talavera[1,2]*

[1]Laboratory of Ion Channel Research and TRP Research Platform Leuven (TRPLe), Department of Cellular and Molecular Medicine, KU Leuven, Leuven, Belgium; [2]VIB Center for Brain & Disease Research, Leuven, Belgium; [3]Center for Microscopy and Molecular Imaging (CMMI), Laboratory of Microscopy, Université Libre de Bruxelles, Gosselies, Belgium

**Abstract** The cation channel TRPA1 transduces a myriad of noxious chemical stimuli into nociceptor electrical excitation and neuropeptide release, leading to pain and neurogenic inflammation. Despite emergent evidence that TRPA1 is regulated by the membrane environment, it remains unknown whether this channel localizes in membrane microdomains or whether it interacts with cholesterol. Using total internal reflection fluorescence microscopy and density gradient centrifugation we found that mouse TRPA1 localizes preferably into cholesterol-rich domains and functional experiments revealed that cholesterol depletion decreases channel sensitivity to chemical agonists. Moreover, we identified two structural motifs in transmembrane segments 2 and 4 involved in mTRPA1-cholesterol interactions that are necessary for normal agonist sensitivity and plasma membrane localization. We discuss the impact of such interactions on TRPA1 gating mechanisms, regulation by the lipid environment, and role of this channel in sensory membrane microdomains, all of which helps to understand the puzzling pharmacology and pathophysiology of this channel.
DOI: https://doi.org/10.7554/eLife.46084.001

*For correspondence:
karel.talavera@kuleuven.vib.be

Competing interests: The authors declare that no competing interests exist.

## Introduction

The detection of harmful stimuli such as noxious chemicals and changes in temperature and pressure are fundamental biological processes carried out by numerous cell types through diverse specialized receptors. The $Ca^{2+}$-permeable non-selective cation channel TRPA1 is arguably the most versatile of such receptors expressed in sensory nerve fibers (*Pedersen et al., 2005*; *Kobayashi et al., 2005*; *Ji et al., 2008*; *Paulsen et al., 2015*), skin keratinocytes (*Anand et al., 2008*; *Atoyan et al., 2009*) and airway epithelial cells (*Nassini et al., 2012*). TRPA1 has been implicated in sensory pathophysiology as detector of thermal (*Story et al., 2003*; *Chen et al., 2013*; *Karashima et al., 2009*; *Hamada et al., 2008*; *Moparthi et al., 2016*; *Moparthi et al., 2014*; *Survery et al., 2016*; *Viswanath et al., 2003*; *Kwan et al., 2006*; *Vandewauw et al., 2018*) and mechanical (*Kwan et al., 2006*; *Kindt et al., 2007*; *Zhang et al., 2008*; *Kwan et al., 2009*) stimuli. However, this channel is best known for being activated by an extremely wide variety of noxious chemicals (*Nilius and Appendino, 2013*; *Nilius et al., 2011*; *Zygmunt and Högestätt, 2014*; *Startek et al., 2019b*). For instance, TRPA1 is activated by electrophilic compounds such as allyl isothiocyanate (AITC) and cinnamaldehyde (*Bandell et al., 2004*; *Bautista et al., 2005*; *Hinman et al., 2006*; *Macpherson et al., 2007*). This channel is also activated by numerous non-electrophilic compounds, some of which have

been proposed to act by inducing mechanical disturbances in the plasma membrane, for example bacterial lipopolysaccharides (LPS) (*Meseguer et al., 2014*; *Soldano et al., 2016*; *Startek et al., 2018*), trinitrophenol and chlorpromazine (*Hill and Schaefer, 2007*). Thus, it can be hypothesized that the function of TRPA1 can be regulated by the surrounding lipid environment (*Startek et al., 2019a*). It was recently reported that the responses of rat trigeminal neurons to the electrophilic agonists AITC and formaldehyde are significantly impaired by pretreatment with the lipid raft desta-bilizer sphingomyelinase (SMase), or with the cholesterol scavenger methyl β-cyclodextrin (MCD) (*Sághy et al., 2015*). Furthermore, a carboxamido-steroid shown to disrupts lipid rafts reduced TRPA1-mediated $Ca^{2+}$ influx in CHO cells transfected with the human isoform and in rat sensory neu-rons (*Sághy et al., 2018*). However, it is currently unknown whether TRPA1 is localized in specific domains of the plasma membrane, or whether cholesterol interacts with specific residues of TRPA1. In this study, we show that mouse TRPA1 is located preferably in cholesterol-rich domains and iden-tify cholesterol recognition amino acid consensus (CRAC) motifs in the TM2 and TM4 segments that are implicated in the attenuation of chemical activation of mTRPA1 by cholesterol-depleting agents.

## Results

### mTRPA1 is localized in lipid rafts

To determine TRPA1 localization in cellular membranes we stained HEK293T cells transfected with the mTRPA1-mCherry construct with the Vybrant Alexa Fluor 488 Lipid Raft Labeling Kit. With the use of total internal reflection fluorescence (TIRF) microscopy we found a population of highly mobile vesicular structures near the cellular membrane containing both mTRPA1-mCherry and the lipid raft marker cholera toxin B (*Figure 1A and B*, *Video 1*), as well as static areas where both fluorescent probes were colocalized at the membrane. Since excitation and emission measurements were sequential, we observed dynamic green and red structures with red fluorescence preceding the green (*Figure 1B*). These dual structures maintained their colors over time and moved together, indicating colocalization (*Figure 1B*). The colocalization of TRPA1 and lipid raft marker was analyzed as previously described (*Ghosh et al., 2016*), yielding a high dynamic colocalization score of 89% (n = 11).

The introduction of the mCherry tag into the C-terminus did not influence TRPA1 responses to its electrophilic agonist AITC, as the half-maximal effective concentration for wild type (WT) mTRPA1 and mTRPA1-mCherry were not significantly different (*Figure 1—figure supplement 1A and B*). Moreover, channel activation by the non-electrophilic agonist thymol (100 or 300 μM) was also unaf-fected by the mCherry tag (*Figure 1—figure supplement 1C and D*).

To further investigate if TRPA1 segregates into lipid rafts, we compared the distribution of mTRPA1-mCherry to that of flotillin-2, plasma membrane gangliosides and the Na/K-ATPase after detergent-resistant membrane (DRM) purification using an OptiPrep flotation gradient. Both flotillin-2 and membrane gangliosides are markers of lipid rafts (*Bickel et al., 1997*; *Zhao et al., 2011*; *Yu et al., 2011*), whereas Na/K-ATPase localizes mainly outside these domains (*Storey et al., 2007*). We found the highest flotillin-2 concentration in the 30% OptiPrep fractions (fractions 3 and 4; *Figure 2A*), similarly to plasma membrane gangliosides detected with the cholera toxin using dot blot (*Figure 2B*), indicating that these fractions correspond to lipid rafts. TRPA1-mCherry was also enriched in the 30–35% gradients confirming our previous observations from the TIRF microscopy (*Figure 2A,C*). The membrane marker Na/K-ATPase was present in the higher fractions, being highly concentrated in the non-raft fraction 6 (*Figure 2A,C*).

Next, we determined whether TRPA1 localization was affected by decreasing membrane choles-terol levels using methyl-β-cyclodextrine (MCD). A 45 min MCD treatment reduced the partition of mTRPA1-mCherry into lipid rafts (fractions 3 and 4), shifting the channel localization to non-raft frac-tions (*Figure 2D,F*). Flotillin-2 and ganglioside levels were reduced in the DRM (*Figure 2D,E*).

A comparison of mTRPA1-mCherry band intensities (*Figure 2—figure supplement 1A*) revealed a shift of at least one fraction between control and cholesterol depletion conditions (*Figure 2—fig-ure supplement 1D*). Flotillin-2 partition between fractions followed the same trend (*Figure 2—fig-ure supplement 1B*), with an average of 1.33 fraction shift (*Figure 2—figure supplement 1D*).

The partition of the non-raft marker NaK-ATPase remained unchanged (*Figure 2—figure supple-ment 1C*), with the highest partitioning in fraction six before and after DRM disruption.

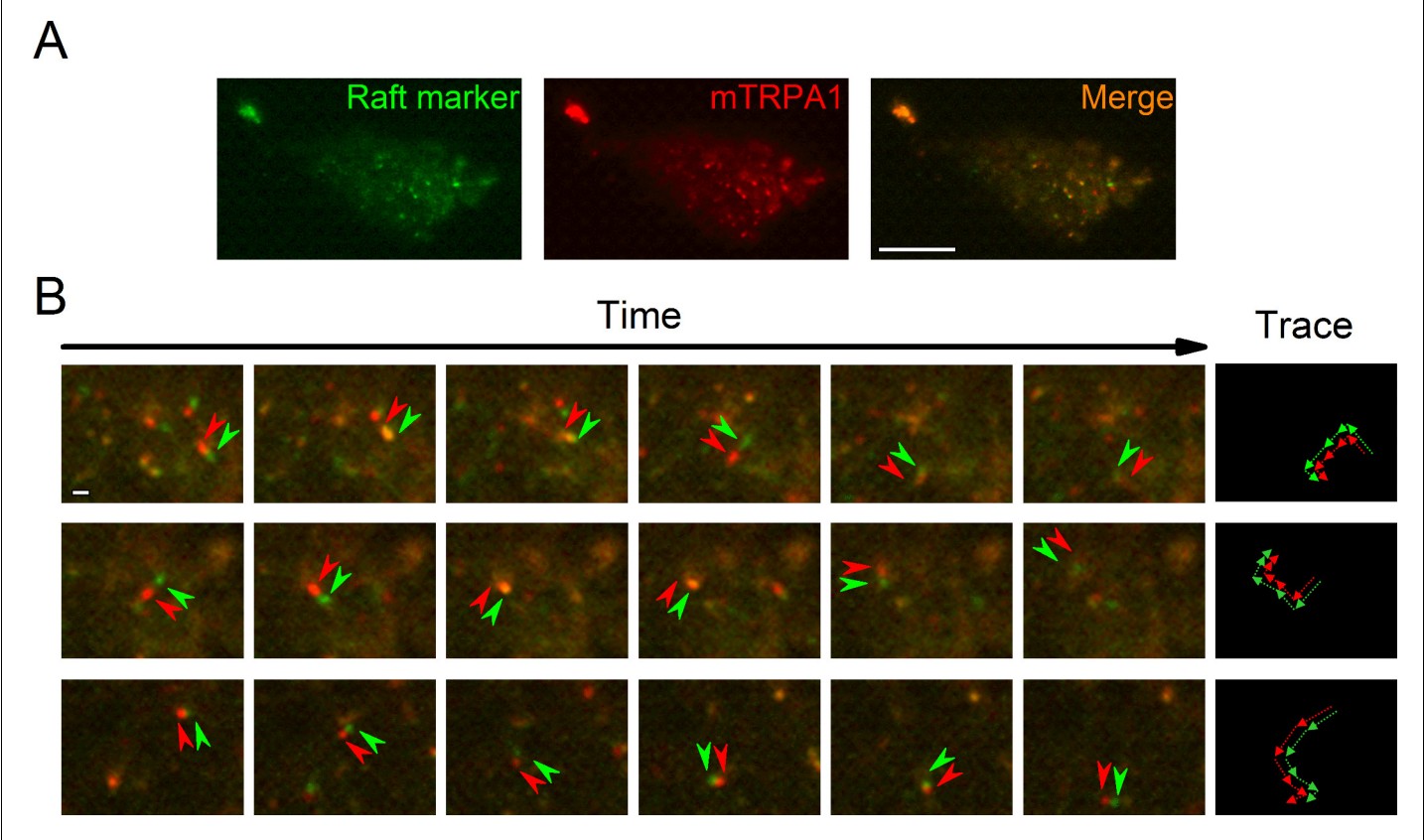

**Figure 1.** TRPA1 colocalizes with the lipid raft marker in the mobile intracellular vesicles near to the plasma membrane. (**A**) Snapshot of TIRF microscopy images recorded in HEK293T cells transfected with the mTRPA1-mCherry construct (red) and stained with Vybrant Alexa Flour 488 lipid raft labeling kit (green). The third panel shows a merged image. Scale bar 10 μm. (**B**) Dual-color TIRF microscopy images at consecutive intervals (every 2 s) displaying the movement of mTRPA1-mCherry (red) along with lipid raft marker (green). Arrowheads indicate traced vesicles whose movements are shown in the rightmost column. Scale bar 2 μm.

DOI: https://doi.org/10.7554/eLife.46084.002

The following source data and figure supplements are available for figure 1:

**Figure supplement 1.** Intracellular calcium responses of mTRPA1 are not influenced by addition of the mCherry tag to channel C-terminus.

DOI: https://doi.org/10.7554/eLife.46084.003

**Figure supplement 1—source data 1.** mTRPA1 response amplitudes and statistical analysis.

DOI: https://doi.org/10.7554/eLife.46084.004

It is important to notice that even after treatment with MCD small amounts of both flotillin-2 and mTRPA1-mCherry were still detected in cholesterol-rich domains. Since the MCD concentration range used in our experimental conditions should remove 30–50% of total cholesterol from the cellular membranes (*Hao et al., 2007*; *Levitt et al., 2009*; *Rodal et al., 1999*; *Pontier et al., 2008*), we still expect formation of cholesterol domains containing raft-associated proteins. Taken together, our TIRF microscopy and gradient separation results suggest that TRPA1 localizes in cholesterol-rich microdomains of the plasma membrane.

## Effect of MCD or SMase treatment on the activity of native and recombinant TRPA1

We determined the effects of lipid raft disruption on the function of native TRPA1 channels using mouse sensory neurons as a well-known cellular model (*Pedersen et al., 2005*; *Nilius and Appendino, 2013*). Ratiometric $Ca^{2+}$ imaging was used to examine the responses of dorsal root ganglion (DRG) neurons to application of the TRPA1 electrophilic agonist AITC or the non-electrophilic agonist thymol, with or without a 30 min pretreatment with 5 mM MCD.

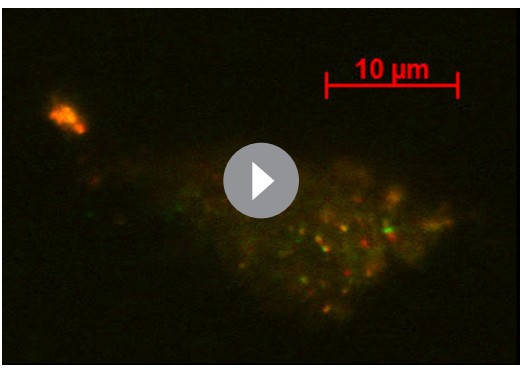

**Video 1.** TIRF microscopy recording of HEKT293 cell transfected with mTRPA1-mCherry (red) and stained with Vybrant Alexa Flour 488 lipid raft labeling kit (green). Time series of images were recorded at intervals of 500 ms and displayed with delay of 200 ms, scale bar 10 μm, digital gain 2, binning 2 × 2.
DOI: https://doi.org/10.7554/eLife.46084.005

In control conditions, 55.3% of neurons responded to 30 μM AITC, with an average increase in intracellular $Ca^{2+}$ concentration ($\Delta[Ca^{2+}]$) of about 500 nM (*Figure 3A,C*). After the MCD pretreatment the percentage of neurons responding to AITC was not different, but the responses were significantly smaller (*Figure 3B,C*). In contrast, MCD did not affect the responses to high $K^+$, suggesting that neither the cell viability nor the excitability were altered. The responses to the TRPV1 agonist capsaicin were 27% smaller in MCD-pretreated cells (*Figure 3C*), confirming the reported modulation of TRPV1 by cholesterol (*Sághy et al., 2015*; *Liu et al., 2006*; *Szoke et al., 2010*). Other experiments showed that MCD also decrease the amplitude of $Ca^{2+}$ responses to 300 μM thymol and confirmed the invariance of the responses to high $K^+$ and the reduction of the effect of capsaicin (*Figure 3D,E,F*).

To further assess whether lipid raft disruption affects TRPA1 activation, we treated DRG neurons with sphingomyelinase (SMase, 50 mUN), which hydrolyzes sphingomyelin (SM), releasing cholesterol and reorganizing membrane domains (*Yu et al., 2005*). SMase reduced the responses to AITC (*Figure 3—figure supplement 1A,B,C*) and thymol, but did not significantly affect the responses to high $K^+$ (*Figure 3—figure supplement 1D, E*).

Similar effects were observed with CHO cells stably expressing mTRPA1 (CHO-mTRPA1). These cells were pretreated with 1, 5 or 10 mM MCD and intracellular $Ca^{2+}$ changes were measured upon application of AITC (100 μM) and thymol (300 μM). As found in DRG neurons, the responses of CHO-mTRPA1 cells to AITC and thymol were reduced by the pretreatment with either MCD or SMase (*Figure 3—figure supplement 2*). These effects were dependent on the MCD and SMase concentrations (*Figure 3—figure supplement 3*). Taken together, these results indicate that, as previously reported for the recombinant human and the native rat TRPA1 isoforms (*Sághy et al., 2015*; *Sághy et al., 2018*), the function of recombinant and native mouse TRPA1 is regulated by cholesterol and the integrity of lipid rafts.

## Molecular determinants of mouse TRPA1 modulation by cholesterol

Since the decrease of cholesterol levels reduced the amplitude of TRPA1-mediated responses, we took a closer look at possible interaction sites of cholesterol in the channel. A few protein motifs have been proposed to directly interact with cholesterol. One of them is the Cholesterol Recognition/interaction Amino acid Consensus (CRAC) motif, which in its first definition consisted of a short linear amino acid sequence composed of a branched apolar Leu or Val residue followed by a 1–5 any residue segment, then a Tyr residue followed by another 1–5 any residue segment and finally a basic Lys or Arg residue: (L/V)-X$_{1-5}$-(Y)-X$_{1-5}$-(K/R) (*Fantini and Barrantes, 2013*; *Fantini et al., 2016*). CRACs are located in transmembrane (TM) domains (N- to C-terminus), usually on the interface between a polar and an apolar environment, and for that reason flanking residues are crucial for stabilizing the TM helix in the bilayer. The aromatic residue on the other hand is located in the apolar region and has a hydroxyl group capable of forming hydrogen bonds. Recently, an extended definition of the CRAC motif was introduced, following the discovery that other aromatic residues such as Phe or Trp can interact with cholesterol (*Fantini et al., 2016*). Thus, the CRAC motif can now be described with the sequence: (L/V)-X$_{1-5}$-(Y/F/W)-X$_{1-5}$-(K/R). Interactions between the CRAC motif and cholesterol are reinforced by the shape of the lipid molecule, which reveals two distinct surfaces that could be defined as a smooth face (α) and a rough face (β) (*Fantini and Barrantes, 2013*; *Fantini et al., 2016*). It has been proposed that branched amino acids such as Leu, Val or Ile can intrude the cholesterol β face via van der Waals interactions (*Fantini and Barrantes, 2013*). Furthermore, the aromatic residue (Y/F/W) present in the CRAC motif can form stable CH-π interactions

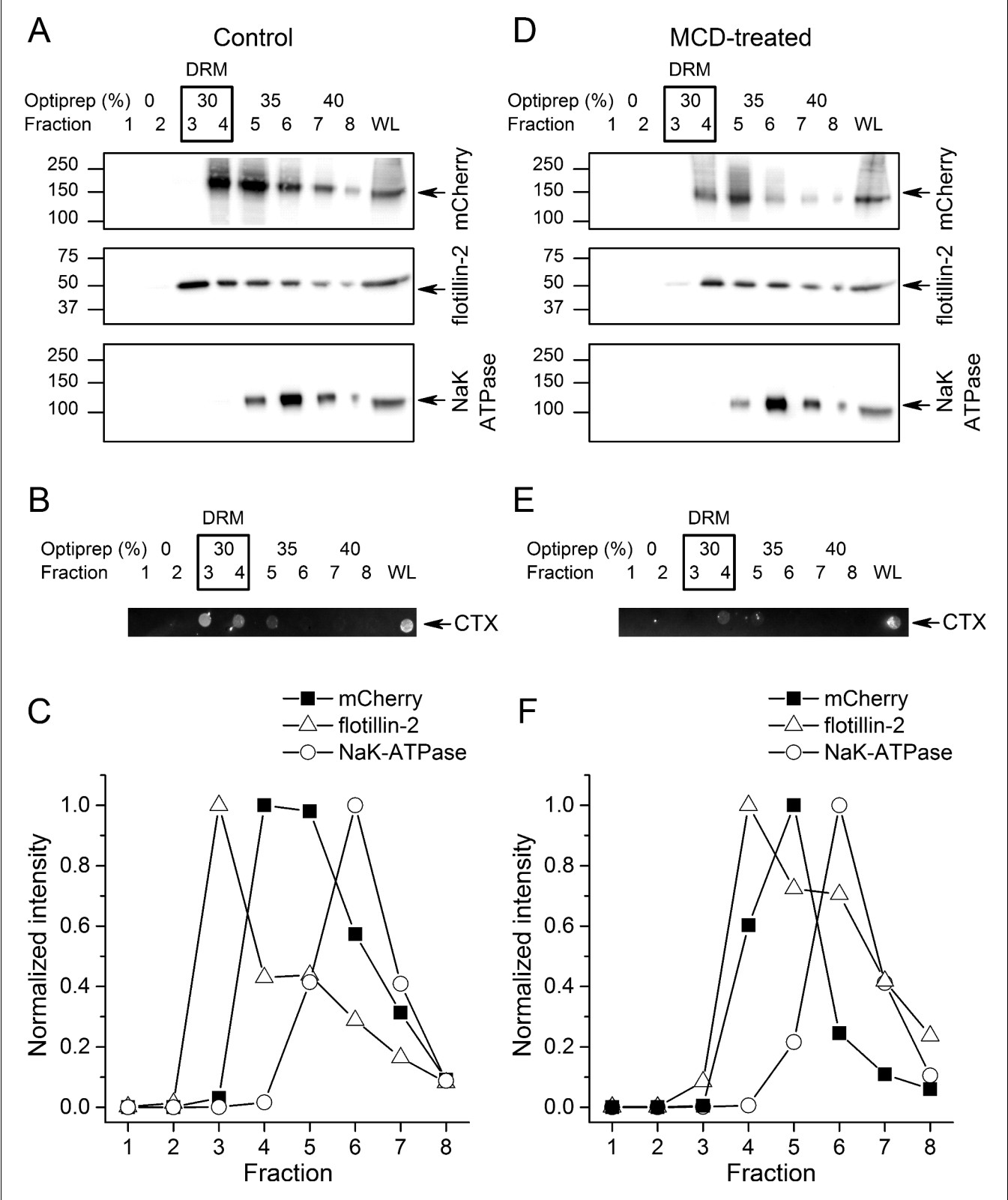

**Figure 2.** TRPA1 localizes in cholesterol-rich domains. (**A**) Immunoblots of different fractions obtained after detergent-free lipid raft preparation presenting mTRPA1-mCherry immunoreactivity associated with detergent-resistant membrane (DRM) fractions and whole cell lysate (WL) in HEK293T-mTRPA1-mCherry cells (top panel); immunoreactivity against endogenous lipid raft marker flotillin-2 (middle panel); immunoreactivity against NaK-ATPase as a marker of non-raft membrane protein (bottom panel). (**B**) Dot blot presenting immunoreactivity associated with fluorescent cholera toxin

*Figure 2 continued on next page*

**Figure 2 continued**

subunit B (CT-B). (**C**) Quantitative analysis of density gradients for mCherry, flotillin-2 and NaK-ATPase. (**D**) Cholesterol depletion of HEK293T-mTRPA1-mCherry cells by treatment with 10 mM MCD interrupts association of TRPA1 (top panel) and flotillin-2 (middle panel) with the detergent-resistant membrane fractions. NaK-ATPase expression remains similar to control condition. (**E**) Dot blot presenting deficiency of fluorescent cholera toxin subunit B binding after lipid raft disruption. (**F**) Quantitative analysis of density gradients for mCherry, flotillin-2 and NaK-ATPase after MCD treatment reveals a shift of mTRPA1-mCherry and flotillin-2 to higher density gradients. Equal volumes of each collected fraction were determined by SDS-PAGE electrophoresis and blotted with specific anti-mCherry, anti-flotillin-2 or anti-NaK-ATPase antibodies.

DOI: https://doi.org/10.7554/eLife.46084.006

The following source data and figure supplements are available for figure 2:

**Figure supplement 1.** Cholesterol depletion by MCD treatment induces shift of lipid raft proteins into higher density fractions.

DOI: https://doi.org/10.7554/eLife.46084.007

**Figure supplement 1—source data 1.** Protein intensities.

DOI: https://doi.org/10.7554/eLife.46084.008

(*Fantini and Barrantes, 2013*; *Fantini et al., 2016*). Yet, interactions of aromatic residues with the β-face are also possible by intercalation between the aliphatic groups. Since mutations in the CRAC motifs have been shown to strikingly reduce or eliminate protein-cholesterol interactions, we wondered if they could be functionally identified in the mTRPA1 channel sequence. We aligned the TRPA1 amino acid sequences from 24 different mammals (taken from Uniprot; www.uniprot.org) and searched for CRAC motifs in the TM domains and the adjacent five amino acid stretches located at the lipid-water interfaces. We identified five possible CRAC motifs, four of which are located in TM2 and one in TM4 (*Figure 4A,B*). Conservation analysis revealed that among these five motifs CRACs 2 (780-**V**FLSSI**F**GYC**K**-790), 3 (782-**L**SSIF**GYCK**-790), 4 (782-**L**SSIFGYC**K**-790) and 5 (850-**L**LY**L**Q**R**-855) (residue numbering corresponding to mTRPA1) are highly conserved throughout mammalian evolution. The CRACs 2, 3 and 4 result from variations of sequences fitting in the same amino acid region.

To investigate the role of CRAC motifs in TRPA1 function, we compared the concentration-dependent responses to AITC, in control and after cholesterol depletion, of WT mTRPA1 and eight mTRPA1 mutants generated by point mutations to alanine at two essential positions: the apolar residues Leu or Val and the aromatic residues Tyr or Phe. Cells transfected with WT mTRPA1 responded in control conditions to AITC with an effective concentration ($EC_{50}$) of $11.8 \pm 2.5$ μM and a maximal intracellular $[Ca^{2+}]$ increase ($\Delta[Ca^{2+}]_{Max}$) of $0.79 \pm 0.4$ μM. In cells pretreated with MCD the $EC_{50}$ was 5-fold higher and the $\Delta[Ca^{2+}]_{Max}$ was 40% of that obtained in control cells. These results show that the reduction of cholesterol levels decreases the sensitivity of TRPA1 to AITC.

The mutants L768A and F771A (in CRAC 1) displayed $EC_{50}$ values (*Figure 5F*) and $\Delta[Ca^{2+}]_{Max}$ ($0.64 \pm 0.05$ μM and $0.72 \pm 0.05$ μM, respectively) similar to those of WT and the MCD treatment increased the $EC_{50}$ (*Figure 5F*) and reduced $\Delta[Ca^{2+}]_{Max}$ to $0.48 \pm 0.05$ μM and $0.46 \pm 0.08$ μM, respectively. In contrast, mutations V780A and F786A in TM2 (CRAC 2) resulted in $EC_{50}$ values in control conditions that are higher than for WT and the pretreatment with MCD did not further increase the $EC_{50}$ or decrease the $\Delta[Ca^{2+}]_{Max}$ values (*Figure 5B,C,F*). CRAC three mutations L782A and Y788A did not change the sensitivity to AITC in either control or cholesterol-depleted conditions (*Figure 5F*; $\Delta[Ca^{2+}]_{Max} = 0.80 \pm 0.08$ μM and $0.62 \pm 0.07$ μM in control and $0.47 \pm 0.04$ μM and $0.17 \pm 0.05$ μM after MCD, respectively). We discarded a role of the CRAC four motif, because it shares with CRAC two and CRAC three the same starting residue where the mutation (L782A) had no effect in the response to AITC. Mutant L850A in TM4 (CRAC 5) displayed a very large $EC_{50}$ and small $\Delta[Ca^{2+}]_{Max}$ ($0.24 \pm 0.02$ μM) in control conditions in comparison with the WT channel, and the MCD treatment did not affect these values (*Figure 5D,F*; $\Delta[Ca^{2+}]_{Max} = 0.20 \pm 0.02$ μM).

Finally, mutation Y852A resulted in a 60% increase of the $EC_{50}$ (*Figure 5E,F*) and a ~ 50% reduction of the $\Delta[Ca^{2+}]_{Max}$ ($0.37 \pm 0.02$ μM) in control conditions with respect to the corresponding values of the WT channel. As for L850A, the MCD treatment did not affect the sensitivity of the Y852A mutant (*Figure 5E,F*; $\Delta[Ca^{2+}]_{Max} = 0.32 \pm 0.01$ μM). Taken together, these results show that residues V780 and F786 in TM2 and L850 and Y852 in TM4 are crucial in the regulation of TRPA1 activity by cholesterol.

The recent elucidation of the structure of the human TRPA1 by cryo-EM (*Paulsen et al., 2015*) allowed us to construct a model of the mouse TRPA1 to further evaluate cholesterol-TRPA1

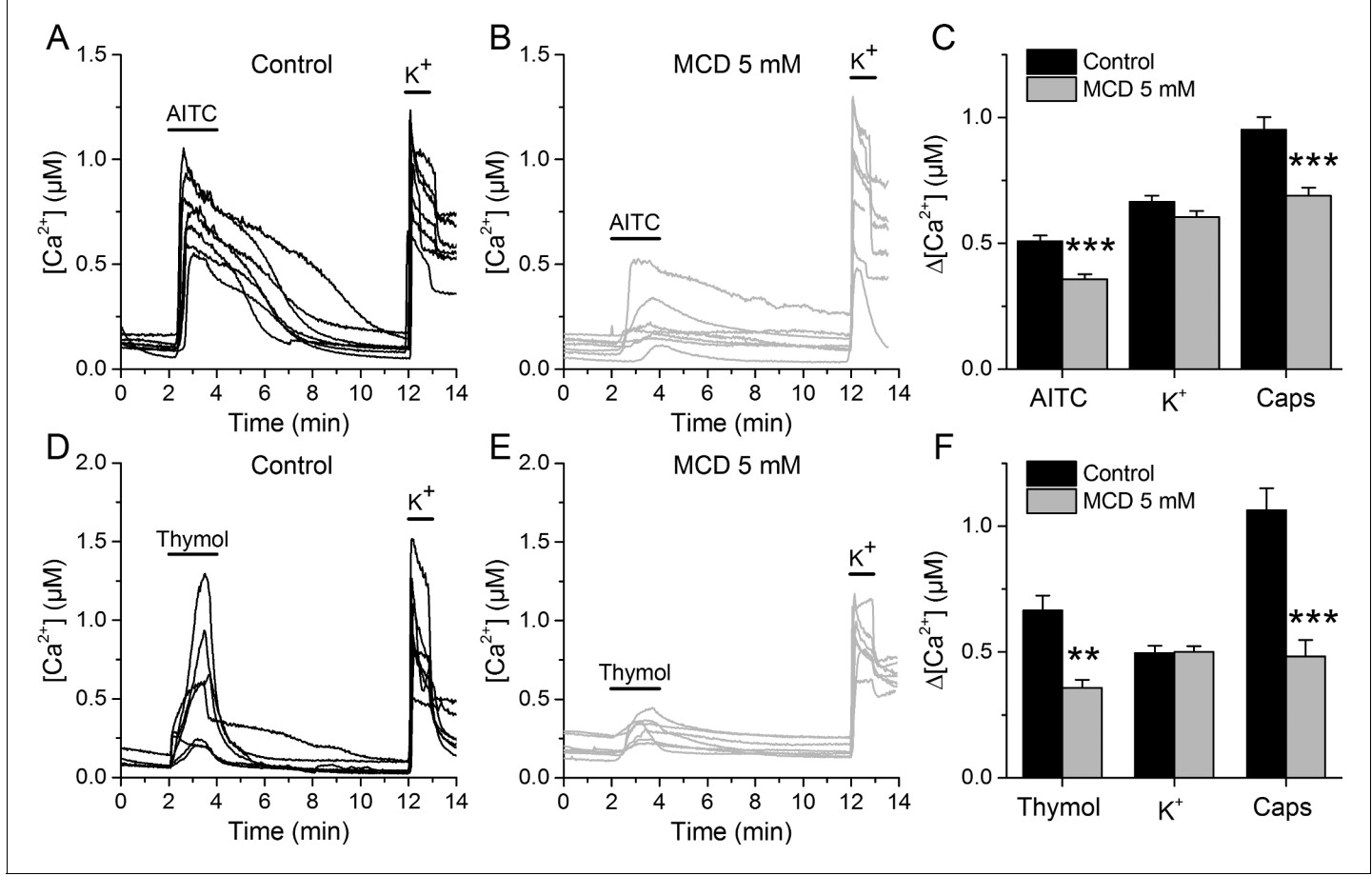

**Figure 3.** TRPA1-mediated responses of mouse sensory neurons are reduced after cholesterol depletion with MCD. (A, B) Examples of [$Ca^{2+}$] responses of mouse DRG neurons to AITC (30 µM) high $K^+$ (50 mM) in control condition (A), n = 340) and after 30 min pretreatment with 5 mM MCD (B), n = 306). (C) Average [$Ca^{2+}$] amplitude change evoked by AITC, capsaicin (Caps) and high $K^+$ in control condition and after MCD (5 mM) treatment. (D, E) Examples of [$Ca^{2+}$] responses of mouse DRG neurons to thymol (300 µM) and high $K^+$ (50 mM) in control condition (D), n = 147) and after MCD treatment (E), n = 122). (F) Average [$Ca^{2+}$] amplitude change evoked by thymol and Caps in control condition and after MCD treatment. The error bars represent the standard error of the mean. *, p<0.05; **, p<0.01; ***, p<0.001; two-tailed Mann-Whitney *U* test.

DOI: https://doi.org/10.7554/eLife.46084.009

The following source data and figure supplements are available for figure 3:

**Source data 1.** mTRPA1 response amplitudes and statistical analysis.

DOI: https://doi.org/10.7554/eLife.46084.015

**Figure supplement 1.** TRPA1-mediated responses of mouse sensory neurons are reduced after lipid raft disruption with SMase.

DOI: https://doi.org/10.7554/eLife.46084.010

**Figure supplement 1—source data 1.** mTRPA1 response amplitudes and statistical analysis.

DOI: https://doi.org/10.7554/eLife.46084.011

**Figure supplement 2.** CHO-mTRPA1 responses to AITC and thymol are reduced after MCD or SMase treatment.

DOI: https://doi.org/10.7554/eLife.46084.012

**Figure supplement 3.** Responses of mTRPA1 channels are reduced after MCD or SMase treatment.

DOI: https://doi.org/10.7554/eLife.46084.013

**Figure supplement 3—source data 1.** mTRPA1 response amplitudes and statistical analysis.

DOI: https://doi.org/10.7554/eLife.46084.014

interactions (*Figure 6A,B*). We found that, due to steric hindrance, cholesterol was not able to dock to any of the transmembrane domains of our model structure of the whole mTRPA1 channel. On the other hand, when docking simulations were performed with individual transmembrane domains in vacuum we identified two cholesterol binding sites in TM2 and two other sites in TM4 (*Figure 6C, D*). In TM2, each binding site contains only one of the two residues defining the CRAC two motif,

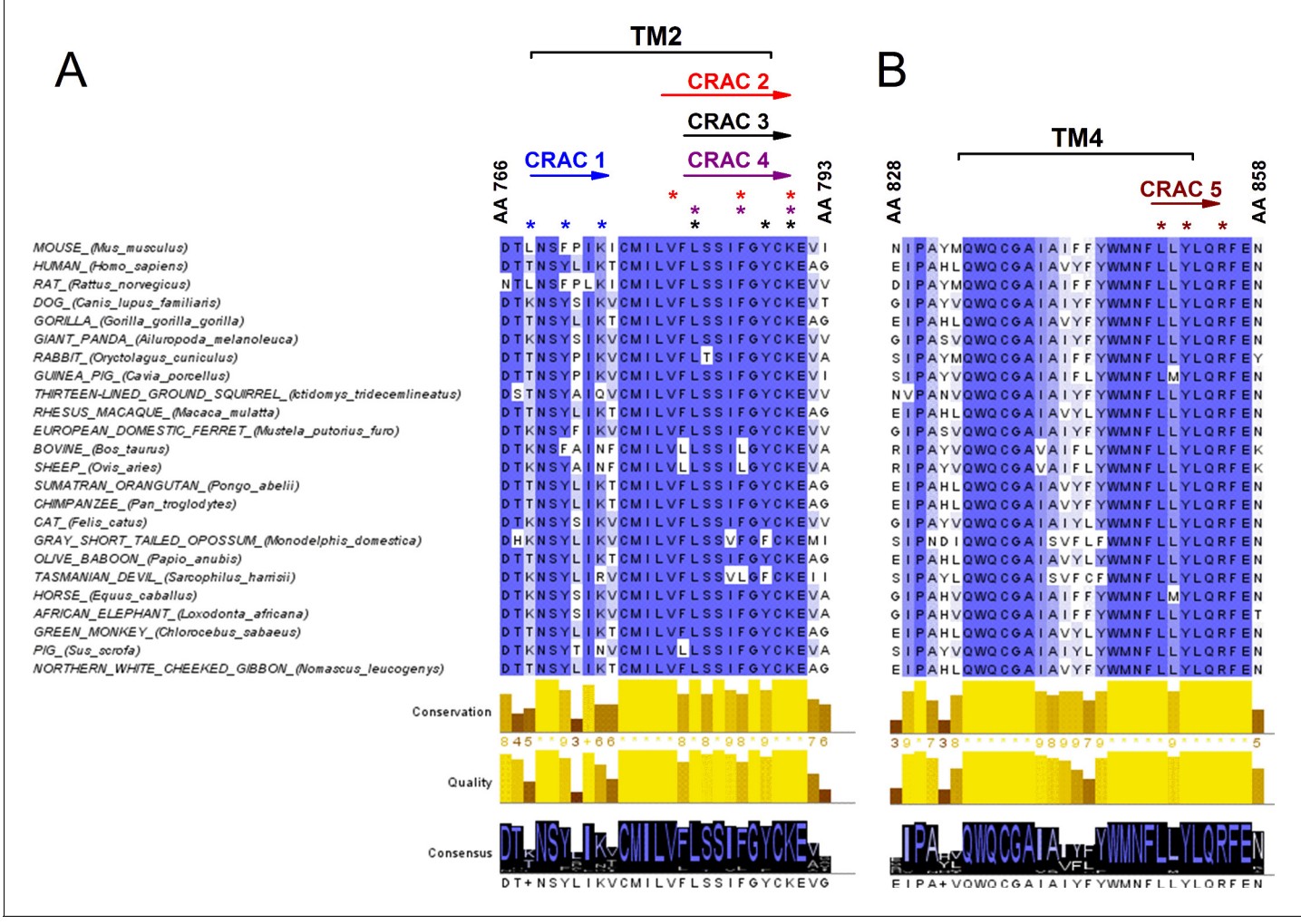

**Figure 4.** Conservation analysis of mammalian TRPA1 reveals highly conserved cholesterol-interaction motifs in TM2 and TM4 domains. (A, B) Sequence alignment of the TM 2 (A) and TM 4 (B) domains of mouse TRPA1 (AA 766–793 and AA 828–858, respectively) with those of 23 other mammals exposes five highly conserved CRAC motifs. Asterisks (*) display localization of residues essential for cholesterol interaction and defined with the algorithm (L/V)-$X_{1-5}$-(Y/F/W)-$X_{1-5}$-(K/R). Conservation, quality and consensus graphs are shown below. Conservation histogram shows the relative similarity score for each column with conserved residues indicated with an asterisk. Columns with mutations where all properties are preserved are marked with a plus. Consensus status is demonstrated as a consensus logo with the scale of the letter in agreement with the conservation of the residues. Multiple sequence alignment was performed with Clustal O in Jalview software.
DOI: https://doi.org/10.7554/eLife.46084.016

either V780 or F786. Similarly, in TM4 the residues L850 and Y852 belong exclusively to either one or the other binding site. Estimations of the interaction energy indicate that in TM2 the binding site containing V780 seems to have higher affinity for cholesterol than the one containing F786 (*Figure 6E*: ΔG for WT site one is more negative than for WT site 2). In contrast, the two binding sites in TM4 are almost energetically indistinguishable (*Figure 6F*: very similar ΔG for WT). In all four dockings cholesterol is found wrapped around the TM2 and TM4 helices. V780 and L850 interact with the hydrocarbon tail of cholesterol, whereas the aromatic residues F786 and Y852 interact with the hydrocarbon rings of the α-face of the molecule. We also found that in silico single mutations of these residues affect the binding energy of their own site but not of the neighboring one (*Figure 6E,F*), further supporting the notion that cholesterol molecules can bind independently to each of the four sites. The calculations also show that the mutations of the amino acids F786 and Y852 produce larger reductions in cholesterol binding energy than the mutations of V780 and L850. This indicates that the aromatic residues have more relative contribution to the affinity for cholesterol of their corresponding binding sites.

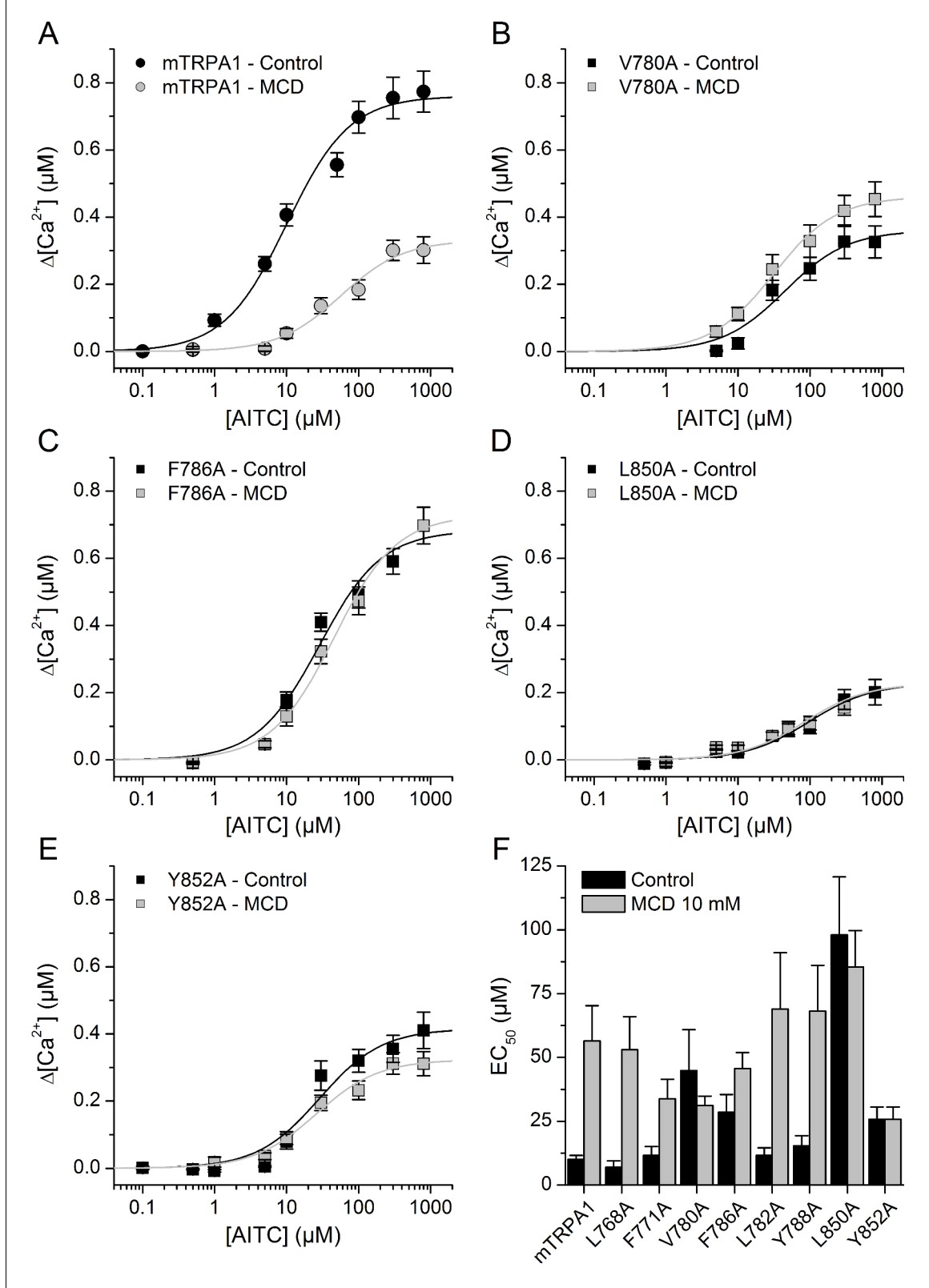

**Figure 5.** Mutation of residues in TM2 and TM4 segments unveils cholesterol binding motifs important for normal mTRPA1 function. (A–D) Dose-response curves derived from [Ca²⁺] measurements (peak fluorescence baseline corrected) in HEK293T cells expressing mTRPA1-mCherry (WT mTRPA1 and single point mutants). $EC_{50}$ curves for (**A**) WT, (**B**) V780A mutant, (**C**) F786A mutant, (**D**) L850A mutant, (**E**) Y852A mutant. Black lines represent fit with Hill equation in the control condition and gray lines after cholesterol removal with MCD. (**F**) AITC $EC_{50}$ concentrations in HEK293T cells expressing

*Figure 5 continued on next page*

*Figure 5 continued*

WT mTRPA1 channel or single-point CRAC mutants in control condition and after cholesterol depletion with 10 mM MCD. Estimation of AITC EC$_{50}$ (± fitting error) for each condition was performed using n > 50 cells from three independent transfection experiments.

DOI: https://doi.org/10.7554/eLife.46084.017

The following source data is available for figure 5:

**Source data 1.** WT mTRPA1 and mutant mTRPA1 response amplitudes and statistical analysis.

DOI: https://doi.org/10.7554/eLife.46084.018

We validated the results obtained in Ca$^{2+}$ imaging experiments by directly assessing the activity of WT and mutant channels using whole-cell patch-clamp recordings in HEK293T cells. AITC (300 µM) enhanced typical TRPA1 currents in cells expressing WT mTRPA1-mCherry or the V780A, F786A, L850A and Y852A mutants (*Figure 7A,B* and *Figure 7—figure supplement 1A,B,C*). All AITC-induced currents were blocked by the TRPA1 antagonist HC030031 (100 µM) (*Figure 7A,B* and *Figure 7—figure supplement 1A,B,C*). The amplitude of the currents elicited by AITC varied across the different channels, with the largest being observed for the WT (*Figure 7C*). These results are fully consistent with the above-described Ca$^{2+}$ imaging measurements, as the current increase strongly correlated with the amplitude of the Ca$^{2+}$ responses (*Figure 7D*; R = 0.94).

To determine whether the lower AITC-induced responses of the mutant channels is due to a reduction of the number of proteins expressed at the membrane, we performed confocal microscopy imaging in cells transfected with WT mTRPA1-mCherry or the mutant channels. WT channels were rather evenly distributed at the membrane in the control condition (*Figure 8A*).

In contrast, cells treated with MCD showed a clustered localization at the plasma membrane, significant amplitude reduction across the cell and strong retention in aggregates underneath the plasma membrane that relates to an ~75% lower amplitude at the plasma membrane (*Figure 8B*). Interestingly, the V780A, F786A, L850A and Y852A mutants displayed altered distribution already in control condition, with features similar to those found for WT in MCD-treated cells (*Figure 8C,E,H*). The MCD treatment failed to reduce the total mCherry signal and the mCherry fraction at the plasma membrane for all mutants except L850A (*Figure 8G,H*). Taken together, these data indicate that mutations in TRPA1 leading to defective interactions with cholesterol also interfere with the pattern of localization of the channel at the plasma membrane.

## Discussion

Several TRP channels have been found to be segregated in lipid rafts. These include members of the TRPC subfamily (TRPC1, TRPC3, TRPC4 and TRPC5) (*Ambudkar et al., 2004*; *Lockwich et al., 2000*; *Brownlow and Sage, 2005*; *Graziani et al., 2006*) and the sensory channels TRPV1 (*Szoke et al., 2010*), TRPV4 (*Kumari et al., 2015*) and TRPM8 (*Morenilla-Palao et al., 2009*). Despite the increasing evidence that TRPA1 activation can be induced by changes in the lipid environment (*Hill and Schaefer, 2007*; *Meseguer et al., 2014*; *Startek et al., 2018*), nothing was known about the localization of this channel in the plasma membrane. Using TIRF microscopy on live cells, we found high colocalization scores for TRPA1 and the lipid raft marker cholera toxin. Ultracentrifugation of Triton-X insoluble fractions revealed a significant co-expression of TRPA1 and the lipid raft marker flotillin-2 in low density fractions, and a concordant shift to higher density fractions upon cholesterol removal with MCD. Thus, we here provide the first evidence that mTRPA1 preferably localizes in lipid rafts. This is notion is further supported by previous (*Sághy et al., 2015*; *Sághy et al., 2018*) and present results showing that lipid raft disruption by MCD or SMase reduces TRPA1-mediated responses in native and heterologous expression systems.

Interestingly, we found weaker effects of the SMase treatment on the responses to AITC in neurons than in CHO cells. SMase induces breakdown of sphingolipids into ceramide, which partially displace cholesterol from the lipid rafts (*Megha and London, 2004*; *Yu et al., 2005*) and leads to the formation of ceramide enriched domains (*De Tullio et al., 2008*). The molar ratio of sphingolipids relative to glycerophospholipids and cholesterol varies across cell types. For instance, sphingolipids are a minor component in erythrocytes, but are particularly abundant in neurons and oligodendrocytes, where they make up 30% of total lipids in myelin sheets (*Ogawa-Goto and Abe, 1998*;

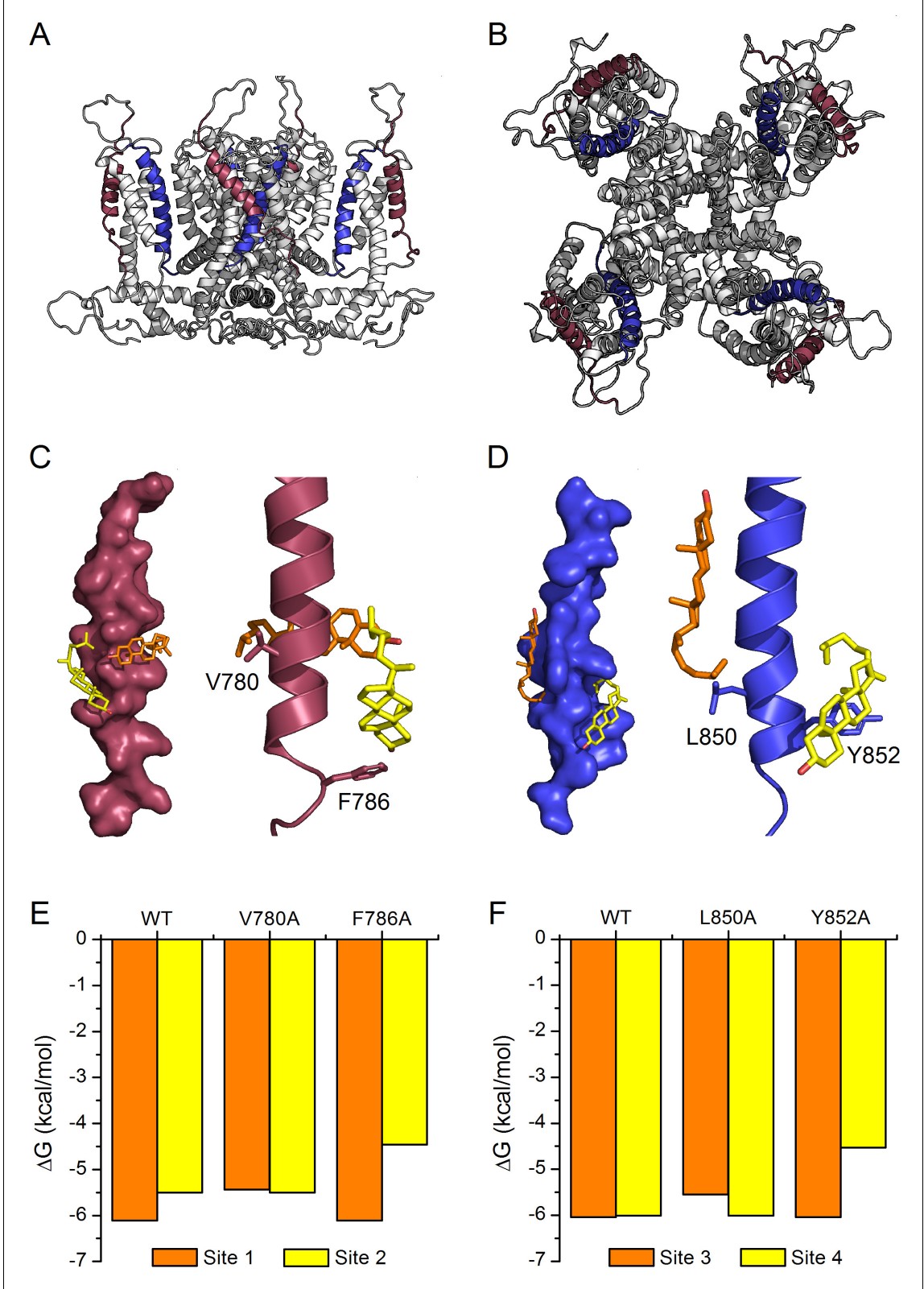

**Figure 6.** Model of cholesterol binding to TM2 and TM4 in TRPA1. (A, B) Model of the mTRPA1 channel (A), side view and (B), top view) presenting localization of TM2 (colored purple) and TM4 (colored blue) segments. (C, D) Two cholesterol molecules (orange and yellow) interact simultaneously with TM2 and TM4 segments (panels C-left and D-left), a zoom-in of residues V780 and F786 (in TM2) and L850 and Y852 (in TM4) showing the close

*Figure 6 continued on next page*

*Figure 6 continued*

proximity of the cholesterol molecules to the side chains (panels C-right and D-right). (**E, F**) Calculated energies of cholesterol binding to the docking sites in the WT and mutant channels.

DOI: https://doi.org/10.7554/eLife.46084.019

*Buccoliero and Futerman, 2003*). Thus, the difference in the effects of SMase we observed between neurons and CHO cells might be related to a differential regulation of ceramide and cholesterol levels in these expression systems. Although not directly comparable, our results seem to be in line with those of Szoke et al. (*Szoke et al., 2010*). They showed that treatment with 10 mUN SMase induced 80% reduction of the amplitude of the responses to capsaicin in CHO cells transfected with TRPV1, but no significant effect on the percentage of cultured trigeminal neurons responsive to capsaicin.

Several mammalian sensory TRP channels (TRPV1, TRPV4, TRPM3 and TRPM8) and the *Drosophila* photoreceptor TRP-like channel (dTRPL) are sensitive to modification of the cholesterol content at the plasma membrane. However, they are distinctly regulated, as cholesterol depletion reduces the activation of TRPV4 (*Lakk et al., 2017*) and dTRPL (*Peters et al., 2017*), but enhances stimulation of TRPM8 (*Morenilla-Palao et al., 2009*) and TRPM3 (*Naylor et al., 2010*). Moreover, although it is clear that TRPV1 function and expression is sensitive to the manipulation of the integrity of lipid rafts (*Szoke et al., 2010*; *Sághy et al., 2015*; *Sághy et al., 2018*), several studies revealed a complex cholesterol-mediated regulation of this channel. For instance, extracellular application of MCD was initially reported not to affect the thermal sensitivity of rat TRPV1 expressed in HEK293 cells (*Liu et al., 2003*), but later shown to reduce the amplitude of currents activated by capsaicin and protons and the amount of TRPV1 protein at the plasma membrane in rat DRG neurons (*Liu et al., 2006*). More recently, extracellular treatment with MCD reduced the effects of capsaicin assessed by $Ca^{2+}$ uptake in CHO cells transfected with rat TRPV1 and the percentage of responding cells in primary cultures of rat trigeminal neurons. However, for still unknown reasons, these findings were replicated when stimulating TRPV1 with resiniferatoxin only in the native expression system (*Szoke et al., 2010*). On the other hand, intracellular application of MCD was reported not to affect capsaicin-induced rat TRPV1 currents in inside-out membrane patches (*Picazo-Juárez et al., 2011*). Finally, cholesterol enrichment by applying extracellular micelles increased the apparent heating threshold for activation and enhanced the thermal sensitivity of rat TRPV1 currents in HEK293 cells (*Liu et al., 2003*). On the other hand, increasing cholesterol by applying a cholesterol-MCD mixture to the inner side of the membrane in excised patches reduced capsaicin-induced currents for the rat isoform and the human variant V585, but not for the human variant I585 (*Picazo-Juárez et al., 2011*).

In summary, it seems clear that the regulation of TRP channels by cholesterol may depend on the channel type, the stimulus, the expression system, the channel species and variant, as well as the membrane side through which cholesterol concentration is manipulated. The later point is particularly critical, given that, to the best of our knowledge, no study on TRP channel regulation has reported how cholesterol flip-flop in the membrane affects concentrations in each leaflet of the bilayer upon intra- or extra-cellular enrichment or depletion. Thus, since we only use mouse TRPA1 and depletion of cholesterol from the outside of the cell, the conclusions of our study should not be extended to other TRPA1 isoforms or to other ways of modifying the membrane cholesterol content.

We found that the treatment with extracellular MCD decreased the maximal response of mTRPA1 channels to AITC, which can be readily explained by the reduced channel expression at the plasma membrane (*Figure 7*), as has been reported for TRPV1 (*Liu et al., 2006*; *Saha et al., 2017*) and TRPV4 (*Lakk et al., 2017*). We also found that, surprisingly, the extracellular MCD treatment strongly reduced the sensitivity of TRPA1 to chemical stimulation with AITC (a 5-fold increased $EC_{50}$). This effect has not been previously reported for any other TRP channel, as for example, the sensitivity of TRPV3 to 2-APB, carvacrol and thymol was not altered by MCD (*Klein et al., 2014*). The decrease in mTRPA1 sensitivity to AITC by MCD cannot be easily explained in terms of impaired functional expression of the channels at the plasma membrane. For this, one would have to consider that the $EC_{50}$ is a decreasing function of the plasma membrane channel density. A simpler

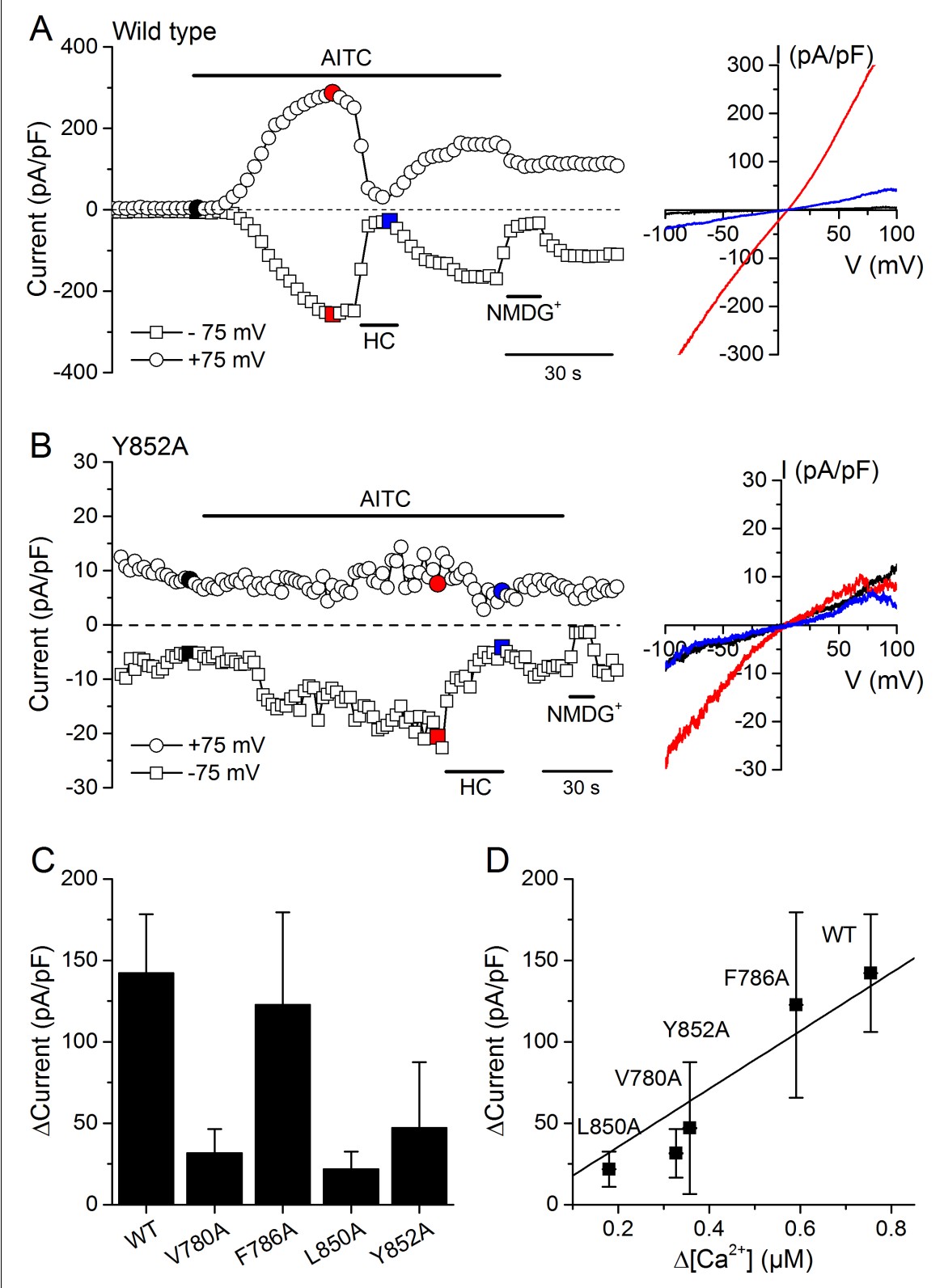

**Figure 7.** Effects of mutations in CRAC motifs in TM2 and TM4 on AITC-evoked TRPA1 currents. (**A, B**) Time course of mTRPA1-mCherry (Wild type) (**A**) or Y852A TRPA1 mutant (**B**) currents response to 300 μM AITC. Currents were blocked using 100 μM HC030031 (HC). The colored data points correspond to the current traces shown on the right. (**C**) Maximal amplitude of currents recorded at −75 mV upon application of AITC (300 μM) in

*Figure 7 continued on next page*

Figure 7 continued

HEK293T cells expressing WT TRPA1, V780A, F786A, L850A or Y852A mutants (n = 4–8). (D) Correlation between the increases in current density and intracellular [Ca$^{2+}$] induced by 300 μM AITC in cells expressing WT TRPA1 or channel mutants.

DOI: https://doi.org/10.7554/eLife.46084.020

The following source data and figure supplement are available for figure 7:

**Source data 1.** Currents of mTRPA1 and channel mutants upon AITC application.

DOI: https://doi.org/10.7554/eLife.46084.022

**Figure supplement 1.** Changes in AITC-evoked TRPA1 currents induced by point mutations in TM2 and TM4 CRAC motifs.

DOI: https://doi.org/10.7554/eLife.46084.021

explanation for an increased EC$_{50}$ is that the MCD treatment partially precluded direct cholesterol-TRPA1 interactions that enhance the binding affinity of AITC and/or the conformational changes leading to channel opening after AITC binding. The existence of such interactions is strongly supported by the identification of two CRAC motifs, in TM2 and TM4, which appear to be crucial for cholesterol-TRPA1 interactions. We identified these motifs by analyzing the effects of mutations in the apolar and aromatic residues, using as criterion that the mutations should render the channel less sensitive to AITC and more resistant to the MCD treatment.

Of note, we found that cholesterol was not able to dock to our model structure of the whole mTRPA1 channel. This is actually not surprising, because we derived our mTRPA1 channel structural model from the human TRPA1 cryo-EM structure that was obtained from a preparation that did not contain cholesterol (*Paulsen et al., 2015*). It is therefore possible that these models are not representative of native structures, or that cholesterol only binds to specific TRPA1 conformations, whose structures are distinct from that obtained by Paulsen et al. Although it is clear that further structural insight and more sophisticated simulations are required to study the dynamics of membrane-cholesterol-TRPA1 interactions, our in silico simulations in vacuum were instrumental in the identification of putative binding sites for cholesterol in the TM2 and TM4 segments. They revealed the possibility of simultaneous interactions with two cholesterol molecules per CRAC motif. Cholesterol docking sites in the TM4 are similar in terms of affinity (ΔG ~ −6 kcal/mol), whereas in TM2 the interaction with V780 appears to be stronger (ΔG = −6.11 kcal/mol) than with F786 (ΔG = −5.5 kcal/mol). The independence between the sites in each TM is further supported by the simulations in which substitution of single CRAC residues by alanine only affected the binding energy at that site.

The interactions between TM2 and TM4 are essential for the function of the voltage-sensing domain in 'classical' voltage-gated ion channels (*Papazian et al., 2002*). However, apart from a few exceptions (e.g., sensing of voltage and menthol by TRPM8 [*Voets et al., 2007*; *Pedretti et al., 2011*; *Pedretti et al., 2009*]), the contributions of these segments to TRP channel gating remain largely unclear. Notably, cholesterol was found to gate K$_V$AP channels by acting on its voltage-sensing domain (*Zheng et al., 2011*) and TM2 and TM4 are also in close contact in the TRPA1 structure (*Paulsen et al., 2015*). Thus, the identification of these segments as cholesterol-dependent regulators of TRPA1 activation supports the idea that the gating mechanisms in voltage-dependent TRP channels are essentially similar to those operating in 'classical' voltage-gated ion channels (*Talavera et al., 2007*; *Voets et al., 2007*). More importantly, this perspective provides for a rather straightforward way to explain how cholesterol depletion induces a decrease in TRPA1 sensitivity for AITC. Indeed, the recent report that not all AITC-sensing elements are located in the N-terminus (*Moparthi et al., 2014*) makes it more understandable that disruption of cholesterol-TM2 and -TM4 interactions result in decreased binding affinity for AITC. It is also possible that such disruption interferes with channel opening after AITC binding. Of note, our finding that the extracellular MCD treatment also reduces the response to thymol indicates for a more general impairment of channel gating and raises the possibility that the mechanisms of activation by other stimuli, such as cold (*Karashima et al., 2009*; *Story et al., 2003*) and mechanically perturbing agents (*Hill and Schaefer, 2007*; *Startek et al., 2018*; *Startek et al., 2019a*), are also affected. Of note, cholesterol depletion did not influence a phenomenon whereby repeated applications of carvacrol induce sensitization of human TRPA1 in the absence of extracellular Ca$^{2+}$ (*Meents et al., 2016*). This agonist-induced sensitization was associated with changes in voltage-dependent gating, but its actual underlying mechanisms and its relevance in the presence of physiological extracellular Ca$^{2+}$ concentrations remain obscure.

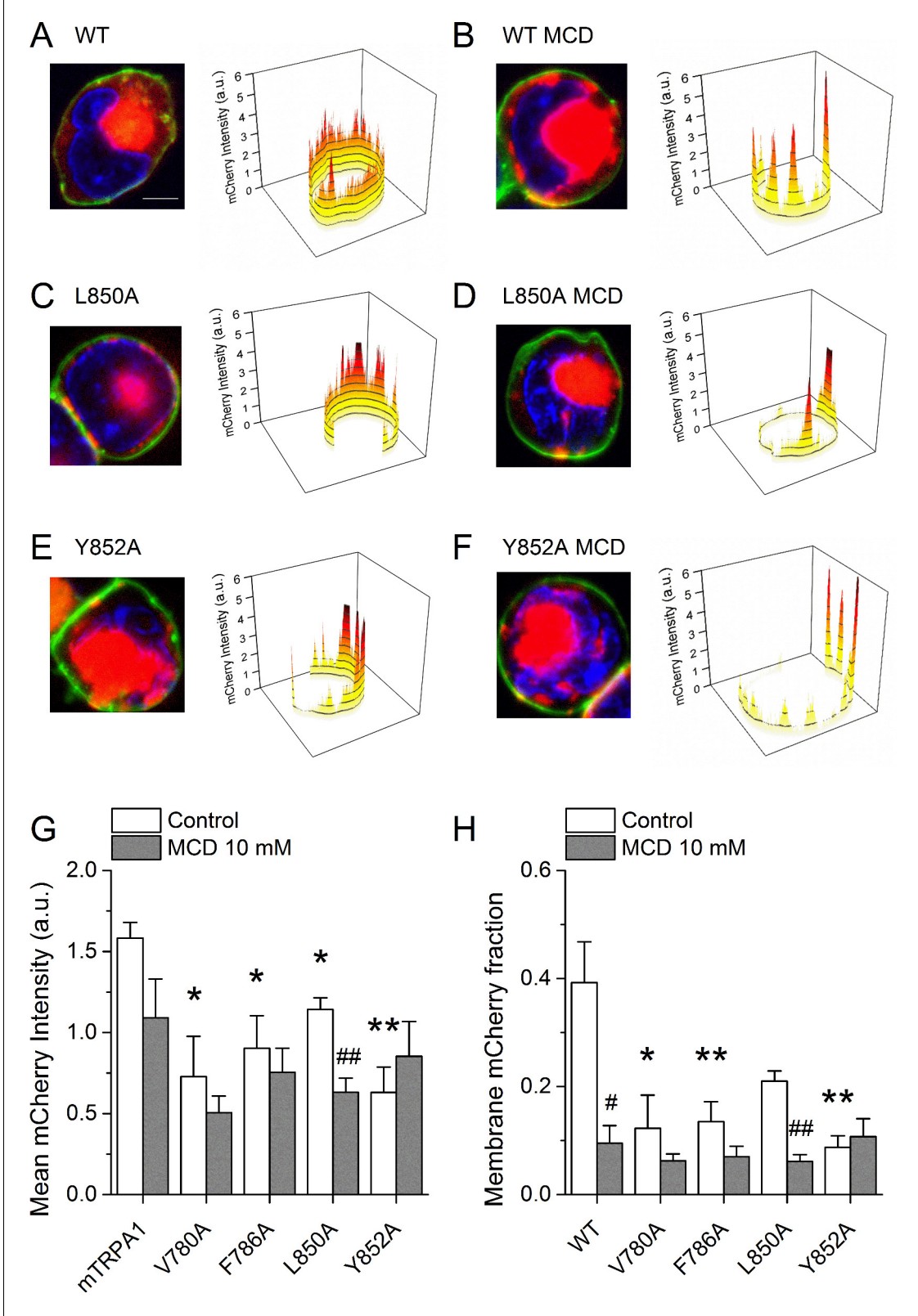

**Figure 8.** Cholesterol-TRPA1 interactions are essential for normal channel distribution in the membrane. (**A–B**) Confocal microscopy images of HEK293T cells expressing mTRPA1-mCherry (red) in the control condition (**A**) and after treatment with MCD (10 mM) (**B**). Expression pattern of mutant L850A TRPA1-mCherry (**C and D**) or Y852A TRPA1-mCherry (**E and F**). Cells were stained with the membrane dye CellBrite (green) and DAPI (blue). Scale bar, 5 μm. Right panels, 3D graph representing total membrane mCherry intensity corresponding the green-stained cell membrane area. (**G**) Mean

*Figure 8 continued on next page*

*Figure 8 continued*

membrane mCherry intensity in HEK293T expressing WT or mutated TRPA1 in control condition and after MCD treatment quantified from confocal microscopy images (n ≥ 5). (**H**) Mean mCherry membrane fractions quantified using custom design software representing distribution pattern of HEK293T expressing WT or mutated TRPA1 in control or after MCD treatment (n ≥ 5). *, $p<0.05$; **, $p<0.01$; #, $p<0.05$; ##, $p<0.01$; two-tailed Mann-Whitney U test.

DOI: https://doi.org/10.7554/eLife.46084.023

The following source data is available for figure 8:

**Source data 1.** Intensities obtained from confocal microscopy and statistical analysis.

DOI: https://doi.org/10.7554/eLife.46084.024

Regarding the role of cholesterol-channel interactions in membrane localization, it was previously observed that mutations in putative cholesterol binding sites in TRPV1 produced proteins that were largely retained intracellularly (*Saha et al., 2017*). However, it was not reported whether those mutations actually resulted in functional channels, nor whether there was a disturbed pattern of expression at the plasma membrane. We found that the integrity of the CRAC motifs in TM2 and TM4 of mTRPA1 is important for proper protein localization, as all four mutants showed decreased and clustered expression at the plasma membrane. This closely resembled the localization pattern observed in cells transfected with the WT channel and treated with MCD, strongly indicating that cholesterol-TRPA1 interactions in TM2 and TM4 are not only relevant for proper chemical sensitivity but also for membrane targeting. To explain the clustered pattern of localization it may be envisaged that upon MCD treatment WT channels display a restricted expression at remaining 'strongholds' of sufficient cholesterol content. Accordingly, the reduced ability of the mutant channels to bind cholesterol may result in a membrane localization restricted to domains with higher cholesterol content.

We conclude that mTRPA1 preferably localizes in lipid rafts and that channel activity and expression pattern is modulated by cholesterol levels. We provide both functional and in silico evidence that cholesterol directly interacts with CRAC motifs present in TM2 and TM4 segments, and that lack of functional interactions with cholesterol changes channel sensitivity to its agonists and influences membrane distribution pattern. Our findings indicate that synthetic expression systems and preparations used for structural analyses not preserving the interactions with cholesterol may be of limited reach in the understanding of channel function. Additionally, the identification of TRPA1 in lipid rafts, together with TRPV1, provides further support to the notion that these channels form signaling complexes in membrane microdomains, through direct protein-protein interactions (*Akopian, 2011*) and/or $Ca^{2+}$-dependent crosstalk (*Patil et al., 2010*). Similar colocalization with bradykinin receptors (*Jeske et al., 2006*) may ensure efficient modulation of TRPA1 (*Bandell et al., 2004*; *Bautista et al., 2006*; *Kádková et al., 2017*). Thus, we propose that lipid rafts constitute scaffolds for a sort of 'esthesiosomes' containing these two key molecular sensors and their close-range modulators.

# Materials and methods

## Cell culture

Human embryonic kidney (HEK293T) cells from the European Collection of Cell Culture (Salisbury, UK) were grown in Dulbecco's modified Eagle's medium (DMEM) containing 10% (v/v) fetal calf serum, 2 mM L-glutamine, two units/ml penicillin and 2 mg/ml streptomycin (Gibco/Invitrogen, Carlsbad, CA, USA) at 37°C in a humidity-controlled incubator with 10% $CO_2$. Cells contamination with mycoplasma species was detected using PlasmoTest - Mycoplasma Detection kit (InvivoGen, Toulouse, France). HEK293T cells were transiently transfected with the pCAGGS-IRES-GFP vector (*Wissenbach et al., 2000*) encoding mouse TRPA1 (*Karashima et al., 2008*), or with the construct corresponding to the TRPA1 (wild type and mutants)-mCherry fusion proteins using Mirus 293 Transfection Reagent (Mirus Corporation, Madison, WI, USA). For most experiments, transfected cells were reseeded after approximately 16 hr on poly-L-lysine-coated (0.1 mg/ml) 25 mm glass coverslips with thickness of 0.16–0.19 mm (Gerhard Menzel GmbH, Braunschweig, Germany) for TIRF experiments or 18 mm glass coverslips with thickness of 0.13–0.16 mm (Gerhard Menzel GmbH) for ratiometric intracellular $Ca^{2+}$ imaging and patch clamp measurements. The mouse TRPA1-mCherry fusion

proteins were constructed by overlap extension polymerase chain reaction (PCR). The final constructs were pCAGGS-mTRPA1-mCherry.

Chinese hamster ovary (CHO-K1) cells from the American Type Culture Collection were grown in DMEM containing 10% fetal bovine serum, 2% glutamax (Gibco/Invitrogen), 1% non-essential amino acids (Invitrogen) and 200 µg/ml penicillin/streptomycin at 37°C in a humidity controlled incubator with 5% $CO_2$. Cells contamination with mycoplasma species was detected using PlasmoTest - Mycoplasma Detection kit (InvivoGen). As TRPA1 expression system we used CHO-K1 cells stably transfected with mouse TRPA1 (CHO-TRPA1) in an inducible system (*Story et al., 2003*).

## Isolation and culture of DRG neurons

Mouse dorsal root ganglion (DRG) neurons were cultured using an adapted protocol of a previously published method (*Descoeur et al., 2011*). Briefly, for each series of experiments 3 to 4 adult mice (10–12 weeks) were killed by cervical dislocation and the lumbosacral (L5-S2) DRGs were bilaterally excised under a dissection microscope. The ganglia were washed in 10% fetal calf serum Neurobasal A medium and later incubated at 37°C in a mix of 1 mg/ml collagenase (Gibco, Grand Island, NY, USA) and 2.5 mg/ml dispase (Gibco) for 45 min. Digested ganglia were gently washed twice with basal medium and mechanically dissociated by mixing with syringes fitted with increasing needle gauges. Neurons were seeded on poly(l-ornithine)/laminin-coated glass chambers (Fluorodish; World Precision Instruments, Hitchin, Hertfordshire, UK) and cultured for 12–18 hr at 37°C in B27-supplemented Neurobasal A medium (Invitrogen) containing 2 ng/ml glial cell line-derived neurotrophic factor (Invitrogen) and 10 ng/ml neurotrophic factor 4 (PeproTech, Rocky Hill, NJ, USA). All protocols were in accordance with the European Community and Belgian Governmental guidelines for the use and care of experimental animals (2010/63/EU, CE Off Jn8L358, LA12110551) and approved by the KU Leuven Ethical Committee Laboratory Animals (Permit Code: In vitro, Prof. Rudi Vennekens).

## Total internal reflection fluorescence microscopy

TIRF microscopy experiments were performed at 25°C using a Krebs solution containing (in mM): 150 NaCl, 6 KCl, 1 $MgCl_2$, 1.5 $CaCl_2$, 10 HEPES, 10 glucose adjusted to pH 7.4. Cells were plated as described above and stained using Vybrant Alexa Fluor 488 Lipid Raft Labeling Kit (Life Technologies, Eugene, OR, USA). Cells were washed with fresh pre-warmed culture medium and incubated with 1 µg/ml of fluorescent cholera toxin subunit B (CT-B) conjugate for 20 min at 37°C in a humidity-controlled incubator. After staining cells were washed twice with warm medium and the glass slide was transferred into the TIRF measurement chamber and filled with Krebs solution. We used a through-the-lens TIRF system built around an inverted Zeiss Axio Observer Z1 microscope equipped with a 100X oil objective with numerical aperture of 1.45 and a Hamamatsu Orca-R2 camera. Fluorescence excitation was performed using a 488 nm (Alexa 488) and 561 nm (mCherry) lasers at 2% of their maximal power, with an incident angle of 65–68°, resulting in an evanescent wave with an expected decay length constant of 80 nm. Time series of images were recorded at intervals of 500 ms, with constant focus guaranteed by use of the Zeiss Definite Focus module. Microscopy images were analyzed using AxioVision 4.8 digital image processing software (Zeiss), ImageJ and Origin 7.0 (OriginLab Corporation, Northampton, MA, USA). To eliminate false-positive colocalization the coupled structures were considered moving together if they stayed together in at least three consecutive frames.

## Detergent-free lipid raft preparation

HEK293T cells were grown in 6-well culture plates until 70–80% confluency and then transfected with mTRPA1-mCherry. Then cells were washed twice with ice-cold $Ca^{2+}$- and $Mg^{2+}$-free PBS (Invitrogen), homogenized in 1% Triton-X lysis buffer (150 mM NaCl, 5 mM DTT, 5 mM EDTA, 25 mM Tris-HCl; pH 7.4) supplemented with a cocktail of protease inhibitors (Sigma-Aldrich, Bornem, Belgium) and solubilized with agitation for 30 min at 4°C (*Morenilla-Palao et al., 2009*). Lysates were passed 10 times via a 21-gauge needle and mixed with 60% OptiPrepTM solution (Axis-Shield, Oslo, Norway) to obtain 40% final concentration (Beckman Instruments Inc, Fullerton, CA, USA). The gradients of 35% and 30% OptiPrepTM solution in 1% Triton-X lysis buffer and lysis buffer alone were gently layered on the top of the 1 ml sample-containing fraction. All steps were performed on ice and reagents were pre-cooled to approximately 4°C. Gradients were then centrifuged at 38000 rpm

for 19 hr at 4°C in a Beckman L-60 Ultracentrifuge and SW55-Ti Beckman rotor. Afterwards, fractions of 500 µl were collected from the top of the tube and further analyzed using Western and dot blot.

## Western blot and dot blot

For Western blot analysis the above described samples were solubilized in 3-fold concentrated sample buffer (240 mM Tris, 30% glycerol, 6% SDS, 3% DTT, and 0.015% bromophenol blue, pH 6.8) by heating to 95°C for 5 min and then subjected to SDS-PAGE using NuPAGE Novex Bis-Tris 4–12% Gels (Life Technologies) and manufacturer's protocol. Separated proteins were transferred to an Immobilon-P PVDF membrane (Millipore, Billerica, MA, USA) and blocked for 1 hr in blocking solution (5% w/v nonfat dry milk in TBS containing 0.1% Tween-20). The membranes were probed with anti-mCherry (1:1000; Abcam, Cambridge, UK), anti-flotillin-2 (1:1000; Sigma-Aldrich), anti-alpha 1 Na-K ATPase (1:5000; Abcam) and anti-β-actin (1:10000; Abcam) antibodies overnight at 4°C. Then, the membranes were washed in TBST and incubated with horseradish peroxidase (HRP)-conjugated secondary antibodies (1:5000; Cell Signaling Technology Inc, Beverly, MA, USA) for 1 hr at room temperature. Immunoreactive complexes were visualized using Amersham ECL Western blotting detection reagent (GE Healthcare, Buckinghamshire, UK) and ChemiDoc MP Imaging System (version 5.01 Beta, Bio-rad Laboratories, Hercules, CA, USA). Results were analyzed using Image Lab Software (version 5.01 Beta, Bio-Rad Laboratories). After visualization membranes were stripped (0.1 M glycine, 0.5% SDS; pH 2.5), washed, blocked and reblotted using the above described procedure.

For dot blot analysis the samples (10 µl) were spotted into a TBST pre-wetted nitrocellulose membrane (Schleicher and Schuell, Keene, NH, USA) using a 96-well Bio-Dot Microfiltration Apparatus (Bio-Rad Laboratories, Richmond, CA, USA). After draining, the membrane was blocked using blocking solution (5% w/v nonfat dry milk in TBST) for 1 hr, incubated for 1 hr at 22°C with cholera toxin subunit B (recombinant) labeled with Alexa Fluor 488 (1:500; Life Technologies) and imaged using ChemiDoc MP Imaging System (version 5.01 Beta). Results were analyzed using Image Lab Software (version 5.01 Beta).

## Disruption of lipid rafts with MCD or SMase

Cells were washed with serum-free culture medium and incubated with different concentrations (1–10 mM) of methyl-β-cyclodextrine (MCD) (Sigma-Aldrich) or sphingomyelinase (SMase) (1–50 mUN) from *Bacillus cereus* (Sigma-Aldrich) for 1 hr at 37°C in a humidity-controlled incubator. After lipid raft disruption cells were washed and used in different procedures.

## Ratiometric intracellular Ca$^{2+}$ imaging

For intracellular Ca$^{2+}$ imaging experiments cells were incubated with 2 µM Fura-2 AM (Biotium, Hayward, CA, USA) for 40 min at 37°C in a humidity-controlled incubator. Fluorescence was measured with alternating excitation at 340 and 380 nm using a monochromator-based imaging system consisting of an MT-10 illumination system (Tokyo, Japan) and Cell$^{M}$ software from Olympus. All experiments were performed using the standard Krebs solution (see above) at 25°C. Data were analyzed and presented as mean ± s.e.m. using Origin 7.0 (OriginLab Corporation).

## Electrophysiological recordings

Whole-cell membrane currents were measured using an EPC-10 patch-clamp amplifier and the software Patchmaster (HEKA electronic, Lambrecht, Germany). Currents were acquired at a sampling rate of 20 kHz, digitally filtered at 2.9 kHz and stored for off-line analysis on a personal computer in an extracellular solution containing (in mM): 150 NaCl, 10 HEPES, 1 MgCl$_2$, pH titrated to 7.4 with NaOH. The pipette solution contained (in mM): 100 Cs-Aspartate, 45 CsCl, 10 EGTA, 10 HEPES, 1 MgCl$_2$, pH titrated to 7.2 with CsOH. Whole-cell currents were elicited by a 200 ms voltage ramp from −120 mV to +120 mV every 2 s from a holding potential of 0 mV. An extracellular solution where NaCl was substituted by NMDG$^{+}$ (N-methyl-D-glucamine) was used to monitor the size of the leak currents.

## Confocal microscopy

HEK293T cells transfected with WT or mutant mTRPA1-mCherry were seeded in glass coverslips and incubated for 45 min with PBS or 10 mM MCD. After treatment, cells were washed and stained with

CellBrite cytoplasmic membrane labeling dye (Biotium, Fremont, CA, USA) and fixed with cold paraformaldehyde. Coverslips were then mounted in glass slides using DAPI-containing mounting solution (VectaShield, Vector Laboratories, Burlingame, CA, USA). The confocal images of labeled cells were collected using the optimal pinhole size for the 63X oil objective of a Zeiss LSM 510 Meta Multiphoton microscope (Carl Zeiss AG, Oberkochen, Germany). To determine the intensity and distribution pattern of WT and mutant mTRPA1-mCherry we used ImageJ (*Schneider et al., 2012*). We selected the region of the membrane based on the observed CellBrite staining. This region of interest (ROI) was then used to generate a new image out of the original mCherry image, by keeping the original intensity values within the ROI and assigning zero intensity everywhere else. This new image was then converted to gray scale to determine the intensity histogram along the membrane (*Figure 8A–F*, right panels). The area covered by the mCherry signal was determined using a custom-made routine that automatically excluded the data points of intensity below the background level (implemented in MATLAB; MathWorks, Natick, MA, US; see the script provided in the file MATLAB-Script F8.xlsx). This area was used to determine the total mCherry intensity within the whole cell (*Figure 8G*), a value that was then used to normalize the intensity of the mCherry signal detected at the membrane (*Figure 8H*).

## Modeling of cholesterol-mTRPA1 interactions

The structure of mouse TRPA1 was built by homology modeling based on the experimentally determined structure of its human homologue (PDB code: 3J9P) (*Paulsen et al., 2015*), using the program Modeller (*Webb and Sali, 2014*). Because we found that cholesterol was not able to dock to our model structure of the whole mTRPA1 channel, the transmembrane segments TM2 (residues 765–793) and TM4 (residues 830–857) were extracted to be separately used as receptors for further docking simulations in vacuum. The software Autodock-Vina (*Trott and Olson, 2010*) was used to predict the possible binding modes of cholesterol to the transmembrane segments mentioned above. In both cases the grid box was set to imbibe the complete helical segment. Cholesterol and the CRAC motifs of TM2 (containing the V780 and F786 residues) and of TM4 (L850 and Y852) were set as flexible for the docking simulations. For each segment, the solutions were then ranked according to their binding energy.

## Data and statistical analysis

The concentration-dependence curves for the stimulatory effect of AITC on mTRPA1 and its mutants were calculated with Hill function fit according to the equation:

$$[Ca^{2+}] = \frac{\text{Max} \times [AITC]^{Hs}}{[AITC]^{Hs} + EC_{50}^{Hs}}$$

where Max stands for the maximal increase in intracellular calcium levels obtained at high concentrations of AITC ([AITC]); $EC_{50}$ for the half effective concentration and $H_S$ for the corresponding Hill coefficient.

If not stated otherwise, the non-parametric Mann-Whitney $U$ test was used to assess statistical significance with the GraphPad Prism software. Results are reported as mean ± s.e.m.; asterisks represent the significance (*p<0.05; **p<0.01; ***p<0.001) and n denotes the sample size.

## Acknowledgements

We thank M Benoit for excellent technical assistance and the members of the LICR for helpful discussions. The CHO-mTRPA1 cell line was kindly provided by Dr. Ardem Patapoutian (The Scripps Research Institute, USA). We would like to thank the Cell Imaging Core facility of the KU Leuven (http://gbiomed.kuleuven.be/english/corefacilities/microscopy/cic/cic.htm) for the use of the confocal microscope. This work was supported by grants of the Research Council of the KU Leuven (GOA/14/011 and C14/18/086) and the Fund for Scientific Research Flanders (FWO: G070212N, G0C7715N and G0D0417N). YAA is a Postdoctoral Fellow of the FWO.

## Additional information

### Funding

| Funder | Grant reference number | Author |
|---|---|---|
| Research Council KU Leuven | GOA/14/11 | Karel Talavera |
| Research Foundation Flanders | G070212N | Karel Talavera |
| Research Foundation Flanders | Postdoctoral Fellowship | Yeranddy A Alpizar |
| Research Council KU Leuven | C14/18/086 | Karel Talavera |
| Research Foundation Flanders | G0C7715N | Karel Talavera |
| Research Foundation Flanders | G0D0417N | Karel Talavera |

The funders had no role in study design, data collection and interpretation, or the decision to submit the work for publication.

### Author contributions

Justyna B Startek, Conceptualization, Formal analysis, Validation, Investigation, Visualization, Writing—original draft, Writing—review and editing; Brett Boonen, Alejandro López-Requena, Ariel Talavera, Formal analysis, Investigation, Visualization, Writing—original draft, Writing—review and editing; Yeranddy A Alpizar, Formal analysis, Investigation, Visualization, Writing—review and editing; Debapriya Ghosh, Formal analysis, Investigation; Nele Van Ranst, Resources; Bernd Nilius, Conceptualization, Writing—review and editing; Thomas Voets, Conceptualization, Supervision, Project administration, Writing—review and editing; Karel Talavera, Conceptualization, Formal analysis, Supervision, Funding acquisition, Visualization, Writing—original draft, Project administration, Writing—review and editing

### Author ORCIDs

Justyna B Startek https://orcid.org/0000-0003-1131-1149
Brett Boonen http://orcid.org/0000-0002-5026-3963
Yeranddy A Alpizar http://orcid.org/0000-0003-1959-5393
Thomas Voets http://orcid.org/0000-0001-5526-5821
Karel Talavera https://orcid.org/0000-0002-3124-138X

### Ethics

Animal experimentation: All protocols were in accordance with the European Community and Belgian Governmental guidelines for the use and care of experimental animals (2010/63/EU, CE Off Jn8L358, LA12110551) and approved by the KU Leuven Ethical Committee Laboratory Animals (Permit Code: In vitro, Prof. Rudi Vennekens).

### Decision letter and Author response

Decision letter https://doi.org/10.7554/eLife.46084.028
Author response https://doi.org/10.7554/eLife.46084.029

## Additional files

### Supplementary files

• Source code 1. MATLAB script.
DOI: https://doi.org/10.7554/eLife.46084.025

• Transparent reporting form
DOI: https://doi.org/10.7554/eLife.46084.026

#### Data availability

All data generated or analysed during this study are included in the manuscript and supporting files. Source data has been provided for Figure 1—figure supplement 1, Figure 2—figure supplement 1, Figure 3, Figure 3—figure supplement 1, Figure 3—figure supplement 3, Figure 5, Figure 7 and Figure 8.

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
