## [Decision Letter]

Thank you for submitting your article "TRPA1 function and membrane localization is modulated by direct interactions with cholesterol" for consideration by *eLife*. Your article has been reviewed by three peer reviewers, including László Csanády as the Reviewing Editor and Reviewer #1, and the evaluation has been overseen by Richard Aldrich as the Senior Editor.

The reviewers have discussed the reviews with one another and the Reviewing Editor has drafted this decision to help you prepare a revised submission.

Summary:

Earlier studies have shown that compounds that disrupt lipid rafts also reduce TRPA1 channel-mediated Ca uptake. Your manuscript examines how interactions of the TRPA1 channel with cholesterol affect cellular localization and agonist-sensitivity of the channel. Using TIRF microscopy and membrane fractionation approaches you show that TRPA1 channels preferentially localize to lipid rafts. Cholesterol depletion using methyl-β-cyclodextrine disrupts this colocalization, and reduces TRPA1 total surface expression, as well as maximal agonist-induced TRPA1 whole-cell currents. In addition, cholesterol depletion reduces TRPA1 sensitivity towards the agonist allyl isothiocyanate (AITC). Using sequence analysis you identify two cholesterol interaction (CRAC) motifs, one in TM2 the other in TM4. Mutations of key residues in these motifs affect both channel localization and function in a manner that resembles the effects of cholesterol depletion. On the other hand, cholesterol depletion itself has no further effect on the mutants. These results provide important new insight into the mechanisms by which TRPA1-cholesterol interactions affect TRPA1 function.

Essential revisions:

1) A more comprehensive discussion of literature addressing TRP channel lipid interactions is needed. Both positive and negative correlations between membrane cholesterol content and TRP channel-mediated currents have been reported. Cholesterol depletion did not affect activation of TRPV1 in studies by Liu, Hui and Qin, 2003, or Picazo-Juarez et al., 2011, whereas it reduced TRPV1 currents in two papers by Saghy et al., 2015, 2018. Conversely, cholesterol enrichment inhibited TRPV1 currents in the study by Picazo-Juarez et al., 2011. Although those studies have not targeted TRPA1, such controversies might be relevant also for TRPA1, and should therefore be cited and discussed (e.g., Discussion, second paragraph). See also comment 2 below.

2) The conclusions should be stated in a less general form. First: the flip-flop rate of cholesterol in artificial membranes is slow (time constant ~100 min, Smith and Green, 1974). Do the authors have an idea on the rate of redistribution of cholesterol between the two leaflets of the membrane in a living cell? Do they know whether, within the time frame of their experiments, manipulations of cholesterol content preferentially affect the membrane leaflet facing the side from which MCD is applied? (This might explain some of the above controversies, as cholesterol depletion was performed from the outside in the Saghy et al. and Liu et al., studies, while cholesterol loading was performed from the inside in the Picazo-Juarez study.) Second: some of the reported effects for TRPV1 showed clear species dependence: in the Picazo-Juarez study cholesterol enrichment of the inner leaflet reduced currents for rTRPV1 but not for one of two variants of hTRPV1. (This might explain other parts of the reported controversies, as Saghy et al. studied mTRPV1, whereas Liu et al. and Picazo-Juarez et al. studied rTRPV1.) The present study focused on mTRPA1, and cholesterol was depleted from the outer membrane leaflet. The authors should therefore limit their conclusions to stating that cholesterol depletion from the external leaflet reduces currents for mouse TRPA1.

3) A study by Meents et al., 2016, showed that mTRPA1 sensitization towards the agonist carvacrol is preserved following cholesterol depletion. Although those results do not necessarily contradict the findings of the current manuscript, they should be cited and discussed. As sensitization seems to be a robust feature of mTRPA1 channels, the authors might consider addressing whether sensitization is preserved following cholesterol depletion even in the face of the reduction in absolute current size that they observe here. This would further enhance the scope of the current study, but we leave it to the authors' discretion whether or not such experiments will be included.

4) The effect of SMase (50 mUN) treatment on AITC responses seems much larger in CHO cells (Figure 3—figure supplement 3C) compared to DRG neurons (Figure 3—figure supplement 1B). Can the authors comment on this difference?

5) In Figure 7—figure supplement 1 currents for the F786A mutant seem sensitized for AITC after HC treatment, in contrast to the V780A mutant or to WT (Figure 7A) for which the current does not fully recover following HC removal. Can the authors comment on this difference? Is it possible that the F786A mutant activates so slowly that the channels have not yet reached full activation at the time when HC was applied? Also, it is stated that HC produces reversible block. Based on the WT data "partially reversible" (during the time-frame of the experiments) would seem more appropriate.

6) Were the docking experiments performed in vacuum, or were the TM helices surrounded by a lipid bilayer? Given that the orientation of cholesterol in the membrane is far from random, but rather highly influenced by bilayer geometry, docking experiments performed in the absence of a bilayer might have limited reliability. Also, what is a more reliable estimator of the relative importance of a residue in cholesterol binding? Is it the absolute value of the abstractly calculated binding energy (DG), or the change in that binding energy upon mutation of the side chain (DDG)? E.g., for the TM2 site, DG is slightly larger for Site 1, but DDG is by far larger for Site 2. In TM4 DG is identical for Sites 3 and 4, but DDG is far larger for Site 4.

---

## [Author Response]

Essential revisions:1) A more comprehensive discussion of literature addressing TRP channel lipid interactions is needed. Both positive and negative correlations between membrane cholesterol content and TRP channel-mediated currents have been reported. Cholesterol depletion did not affect activation of TRPV1 in studies by Liu, Hui and Qin, 2003, or Picazo-Juarez et al., 2011, whereas it reduced TRPV1 currents in two papers by Saghy et al., 2015, 2018. Conversely, cholesterol enrichment inhibited TRPV1 currents in the study by Picazo-Juarez et al., 2011. Although those studies have not targeted TRPA1, such controversies might be relevant also for TRPA1, and should therefore be cited and discussed (e.g., Discussion, second paragraph). See also comment 2 below.

We agree with the evaluators in that according to the literature the regulation of TRPV1 by cholesterol seems to be very complex. That is why we did not want to address this subject in detail in the first version of our paper.

In addition to the possible explanations for the apparently paradoxical findings on TRPV1 given by the reviewers in this and the following point, it must be noted that some studies have looked at the responses to capsaicin and others to the sensitivity to heat, making it difficult if not impossible to compare directly the effects of cholesterol manipulations. However, we kindly request the evaluators to appreciate that our manuscript is not an appropriate scenario to clear out the intricacies of the modulation of TRPV1 by cholesterol. We do not provide any new data on this channel since this was not our objective. Nevertheless, to comply with the request we now make extensive reference to the literature on TRPV1 (please, see Discussion, third and fourth paragraphs).

2) The conclusions should be stated in a less general form. First: the flip-flop rate of cholesterol in artificial membranes is slow (time constant ~100 min, Smith and Green, 1974). Do the authors have an idea on the rate of redistribution of cholesterol between the two leaflets of the membrane in a living cell? Do they know whether, within the time frame of their experiments, manipulations of cholesterol content preferentially affect the membrane leaflet facing the side from which MCD is applied? (This might explain some of the above controversies, as cholesterol depletion was performed from the outside in the Saghy et al. and Liu et al., studies, while cholesterol loading was performed from the inside in the Picazo-Juarez study.) Second: some of the reported effects for TRPV1 showed clear species dependence: in the Picazo-Juarez study cholesterol enrichment of the inner leaflet reduced currents for rTRPV1 but not for one of two variants of hTRPV1. (This might explain other parts of the reported controversies, as Saghy et al. studied mTRPV1, whereas Liu et al. and Picazo-Juarez et al. studied rTRPV1.) The present study focused on mTRPA1, and cholesterol was depleted from the outer membrane leaflet. The authors should therefore limit their conclusions to stating that cholesterol depletion from the external leaflet reduces currents for mouse TRPA1.

In addition to the estimate provided by the reviewers (Smith and Green, 1974) we found references for cholesterol diffusion across bilayers on time scales of seconds or less (Lange et al., 1981, Muller and Herrmann, 2002, Gu et al., 2019), minutes (Rodrigueza et al., 1995, Schroeder et al., 1996, Haynes et al., 2000, Leventis and Silvius, 2001), or hours (Brasaemle et al., 1988, Rodrigueza et al., 1995). However, please note that we do not have the technical capabilities to determine neither the rate of cholesterol flip-flop nor its influence on cholesterol concentrations in each membrane leaflets upon application of MCD. Nevertheless, it seems reasonable to assume that this treatment reduces the total concentration of cholesterol in the membrane and therefore the availability of this molecule to bind to the CRAC domains we identify.

In order to comply with this request, we substituted in the text the term “cholesterol depletion” for the more technically correct “treatment with extracellular MCD” or similar. We also added a statement that makes clear the limits of our conclusions, which reads as follows:

“In summary, it seems clear that the regulation of TRP channels by cholesterol may depend on the channel type, the stimulus, the expression system, the channel species and variant, and the membrane side through which cholesterol concentration is manipulated. […] Thus, since we use only mouse TRPA1 and depletion of cholesterol from the outside of the cell, the conclusions of our study should not be extended to other TRPA1 isoforms or to other ways of modifying the membrane cholesterol content.”

3) A study by Meents et al., 2016, showed that mTRPA1 sensitization towards the agonist carvacrol is preserved following cholesterol depletion. Although those results do not necessarily contradict the findings of the current manuscript, they should be cited and discussed. As sensitization seems to be a robust feature of mTRPA1 channels, the authors might consider addressing whether sensitization is preserved following cholesterol depletion even in the face of the reduction in absolute current size that they observe here. This would further enhance the scope of the current study, but we leave it to the authors' discretion whether or not such experiments will be included.

As indicated by the reviewers the results of Meents et al., 2016, are not at odds with our findings or conclusions because they did not report on the effects of cholesterol depletion on the absolute current amplitudes. We thank for this suggestion, but there are several elements that decrease our enthusiasm to perform and include the suggested experiments in the present study. First, Meents et al. tested the effect of cholesterol depletion on the sensitization of human TRPA1 and we studied the mouse isoform. Second, the sensitization was observed only in the absence of extracellular Ca^2+^ (see also Meents et al., PLoS One, 2017 and Raisinghani et al., Am J Physiol Cell Physiol, 2011). Third, they actually found no effect of cholesterol depletion on the sensitization pattern. Finally, the mechanism underlying agonist-induced sensitization is far from being well understood. Thus, we would have to study this still obscure phenomenon on mouse TRPA1, in non-physiological conditions (Ca^2+^ free), to perhaps find no effects of cholesterol depletion. As we mentioned in the cover letter of the initial submission our work generates many new questions, but we think that they should be addressed in future studies. Nevertheless, we now make reference to the results obtained by Meents et al., 2016, on the human TRPA1 isoform (please, see Discussion, end of seventh paragraph).

4) The effect of SMase (50 mUN) treatment on AITC responses seems much larger in CHO cells (Figure 3—figure supplement 3C) compared to DRG neurons (Figure 3—figure supplement 1B). Can the authors comment on this difference?

SMase induces breakdown of sphingolipids into ceramide, which has been shown to partially displace cholesterol from the lipid rafts (Megha and London, 2004; Yu et al., 2005) and to lead to the formation of ceramide enriched domains (De Tullio et al., 2008). The molar ratio of sphingolipids relative to glycerophospholipids and cholesterol varies across cell types. For instance, sphingolipids are a minor component in erythrocytes, but are particularly abundant in neurons and oligodendrocytes, where they make up 30% of total lipids in myelin sheets (Buccoliero et al., 2003; Ogawa-Goto and Abe, 1998). Thus, the weaker effects of the SMase treatment on the responses to AITC in neurons compared to those found in CHO cells might be related to a differential regulation of ceramide and cholesterol levels in these cells. Although not directly comparable, our results seem to be in line with those of Szoke et al., 2010. They showed that treatment with 10 mUN SMase induced 80% reduction of the amplitude of the responses to capsaicin in CHO cells transfected with TRPV1, but no significant effect on the percentage of cultured trigeminal neurons responsive to capsaicin. We refer to this issue in the revised version of the manuscript (please, see Discussion, second paragraph).

5) In Figure 7—figure supplement 1 currents for the F786A mutant seem sensitized for AITC after HC treatment, in contrast to the V780A mutant or to WT (Figure 7A) for which the current does not fully recover following HC removal. Can the authors comment on this difference? Is it possible that the F786A mutant activates so slowly that the channels have not yet reached full activation at the time when HC was applied? Also, it is stated that HC produces reversible block. Based on the WT data "partially reversible" (during the time-frame of the experiments) would seem more appropriate.

Indeed, in this experiment the TRPA1 blocker was applied before AITC produced its full effect. If one follows the envelope of the time course of the current amplitude it is clear that the effect of the blocker interrupts the increase in current induced by AITC.

6) Were the docking experiments performed in vacuum, or were the TM helices surrounded by a lipid bilayer? Given that the orientation of cholesterol in the membrane is far from random, but rather highly influenced by bilayer geometry, docking experiments performed in the absence of a bilayer might have limited reliability.

We are fully aware that any model has limited reliability. The calculations were indeed performed in vacuum. Please, note that docking simulations are performed to determine whether there is a theoretical possibility of binding to specific sites on the protein. To test whether cholesterol is able to migrate from the lipid bilayer to the putative binding sites on TRPA1 would require a much higher computational power (which is not readily available to us). Moreover, one would need to use a model of the whole channel structure that allows docking of cholesterol. However, we found that cholesterol was not able to dock to our model structure of the whole mTRPA1 channel. This is rather inconvenient, but actually not surprising, because we derived our mTRPA1 channel structural model from a human TRPA1 cryo-EM structure that was obtained from a preparation that did not contain cholesterol (Paulsen et al., 2015). It is therefore possible that these models are not representative of native structures, or that cholesterol only binds to specific TRPA1 conformations, whose structures are distinct from that obtained by Paulsen et al. Nevertheless, our simulations were instrumental in the identification of putative binding sites for cholesterol in the TM2 and TM4 segments, which was the objective we pursued. It is clear that further structural insight and more sophisticated simulations are required to study the dynamics of membrane-cholesterol-TRPA1 interactions. Thanks to the reviewer’s comment we now make these aspects clear in the revised version of the manuscript, as follows:

- In the Results we now write: “We found that, due to steric hindrance, cholesterol was not able to dock to any of the transmembrane domains of our model structure of the whole mTRPA1 channel. On the other hand, when docking simulations were performed with individual transmembrane domains in vacuum we identified two cholesterol binding sites in TM2 and two other sites in TM4 (Figure 6C, D).”

- In the Discussion we now write: *“*Of note, we found that cholesterol was not able to dock to our model structure of the whole mTRPA1 channel. […] Although it is clear that further structural insight and more sophisticated simulations are required to study the dynamics of membrane-cholesterol-TRPA1 interactions, our in silico simulations in vacuum were instrumental in the identification of putative binding sites for cholesterol in the TM2 and TM4 segments.”

- In the Materials and methods we now write: “Because we found that cholesterol was not able to dock to our model structure of the whole mTRPA1 channel, the transmembrane segments TM2 (residues 765 – 793) and TM4 (residues 830 – 857) were extracted to be separately used as receptors for further docking simulations in vacuum.”

Also, what is a more reliable estimator of the relative importance of a residue in cholesterol binding? Is it the absolute value of the abstractly calculated binding energy (DG), or the change in that binding energy upon mutation of the side chain (DDG)? E.g., for the TM2 site, DG is slightly larger for Site 1, but DDG is by far larger for Site 2. In TM4 DG is identical for Sites 3 and 4, but DDG is far larger for Site 4.

The most reliable estimator of the relative importance of a particular residue in a binding site is the difference in binding energy calculated for the corresponding mutant and the binding energy calculated for the wild type. On the other hand, the absolute binding energy calculated for the wild type channel gives an indication of how strong the binding of cholesterol to each site is. Please, note that these results are not contradictory. To clearly state the relevance of our calculation we added the following text:

“The calculations also show that the mutations of the amino acids F786 and Y852 produce larger reductions in cholesterol binding energy than the mutations of V780 and L850. This indicates that the aromatic residues have more relative contribution to the affinity for cholesterol of their corresponding binding sites.”